# The nature of genetic and environmental susceptibility to multiple sclerosis

**Douglas S. Goodin**[1]*, **Pouya Khankhanian**[2], **Pierre-Antoine Gourraud**[1,3,4], **Nicolas Vince**[3,4]

**1** Department of Neurology, University of California, San Francisco, San Francisco, CA, United States of America, **2** Center for Neuro-Engineering and Therapeutics, University of Pennsylvania, Philadelphia, PA, United States of America, **3** Centre de Recherche en Transplantation et Immunologie UMR 1064, INSERM, Université de Nantes, Nantes, France, **4** Institut de Transplantation Urologie Néphrologie (ITUN), CHU Nantes, Nantes, France

* douglas.goodin@ucsf.edu

**Data Availability Statement:** All relevant data are within the manuscript and its Supporting Information files.

**Funding:** The authors received no specific funding for this work.

## Abstract

### Objective

To understand the nature of genetic and environmental susceptibility to multiple sclerosis (MS) and, by extension, susceptibility to other complex genetic diseases.

### Background

Certain basic epidemiological parameters of MS (e.g., population-prevalence of MS, recurrence-risks for MS in siblings and twins, proportion of women among MS patients, and the time-dependent changes in the sex-ratio) are well-established. In addition, more than 233 genetic-loci have now been identified as being unequivocally MS-associated, including 32 loci within the major histocompatibility complex (*MHC*), and one locus on the *X* chromosome. Despite this recent explosion in genetic associations, however, the association of MS with the *HLA-DRB1\*15:01~HLA-DQB1\*06:02~a1* (*H+*) haplotype has been known for decades.

### Design/Methods

We define the "genetically-susceptible" subset ($G$) to include everyone with any non-zero life-time chance of developing MS. Individuals who have no chance of developing MS, regardless of their environmental experiences, belong to the mutually exclusive "non-susceptible" subset ($G–$). Using these well-established epidemiological parameters, we analyze, mathematically, the implications that these observations have regarding the genetic-susceptibility to MS. In addition, we use the sex-ratio change (observed over a 35-year interval in Canada), to derive the relationship between MS-probability and an increasing likelihood of a sufficient environmental exposure.

### Results

We demonstrate that genetic-susceptibilty is confined to less than 7.3% of populations throughout Europe and North America. Consequently, more than 92.7% of individuals in

**Competing interests:** The authors have declared that no competing interests exist.

these populations have no chance whatsoever of developing MS, regardless of their environmental experiences. Even among carriers of the *HLA-DRB1*15:01~HLA-DQB1*06:02~a1* haplotype, far fewer than 32% can possibly be members the (*G*) subset. Also, despite the current preponderance of women among MS patients, women are less likely to be in the susceptible (*G*) subset and have a higher environmental threshold for developing MS compared to men. Nevertheless, the penetrance of MS in susceptible women is considerably greater than it is in men. Moreover, the response-curves for MS-probability in susceptible individuals increases with an increasing likelihood of a sufficient environmental exposure, especially among women. However, these environmental response-curves plateau at under 50% for women and at a significantly lower level for men.

## Conclusions

The pathogenesis of MS requires both a genetic predisposition and a suitable environmental exposure. Nevertheless, genetic-susceptibility is rare in the population (< 7.3%) and requires specific combinations of non-additive genetic risk-factors. For example, only a minority of carriers of the *HLA-DRB1*15:01~HLA-DQB1*06:02~a1* haplotype are even in the (*G*) subset and, thus, genetic-susceptibility to MS in these carriers must result from the combined effect this haplotype together with the effects of certain other (as yet, unidentified) genetic factors. By itself, this haplotype poses no MS-risk. By contrast, a sufficient environmental exposure (however many events are involved, whenever these events need to act, and whatever these events might be) is common, currently occurring in, at least, 76% of susceptible individuals. In addition, the fact that environmental response-curves plateau well below 50% (especially in men), indicates that disease pathogenesis is partly stochastic. By extension, other diseases, for which monozygotic-twin recurrence-risks greatly exceed the disease-prevalence (e.g., rheumatoid arthritis, diabetes, and celiac disease), must have a similar genetic basis.

## Introduction

The nature of susceptibility to multiple sclerosis (MS) is complex and involves both environmental and genetic factors [1–4]. Recently, considerable progress has been made in our understanding of the genetic basis for susceptibility to MS. Thus, to date, using genome-wide association screens (GWAS), which incorporate large arrays of single nucleotide polymorphisms (*SNPs*) scattered throughout the genome, more than 200 common risk variants (located in diverse genomic regions) have been identified as being MS-associated [5–14]. For example, in the recent GWAS study from the International MS Genetics Consortium [14], 233 *SNPs* (loci) were identified as being associated with MS susceptibility, including 32 loci within the major histocompatibility complex (*MHC*), and one locus identified on the *X*-chromosome. These MS-associated *SNPs* are located within or near to immune-related genes that implicate both the adaptive and innate arms of the immune system. Despite this recent explosion in the number of identified MS-associated regions, however, the association of MS-susceptibility with the *HLA-DRB1*15:01~HLA-DQB1*06:02* haplotype of the human leukocyte antigens (*HLA*), inside the *MHC*, has been known for decades [11, 15–22]. We have recently identified an 11-*SNP* haplotype (*a1*), which adds further specificity to the description of this particular

genetic association [23, 24]. This *SNP*-haplotype spans 0.25 megabases (*mb*) of DNA surrounding the *HLA-DRB1* gene on the short arm of chromosome 6 [23, 24]. It has the most significant association with MS of any *SNP*-haplotype in the genome, and it is tightly linked to the *HLA-DRB1*15:01~HLA-DQB1*06:02* haplotype [23, 24]. For example, 99% of these (*a1*) *SNP*-haplotypes carry this *HLA*-haplotype and, conversely, 99% of these *HLA*-haplotypes carry the (*a1*) *SNP*-haplotype [23, 24]. In the Welcome Trust Case Control Consortium (WTCCC) dataset, the odds ratio (*OR*) for an association the full *HLA-DRB1*15:01~HLA-DQB1*06:02~ a1* haplotype with MS was 3.28 ($p << 10^{-300}$) and similar disease associations for portions of this haplotype have been consistently reported in many other MS populations across Northern Europe and North America [11, 15–24].

Despite the undoubted influence of genetic and environmental factors in MS-pathogenesis, susceptibility to MS might be envisioned in number of different ways. Four examples of disease states, for which we understand, generally, the pathophysiology, can be helpful to highlight some of the issues that might also be involved in MS pathogenesis.

First, sickle cell disease (*SCD*) occurs in ~3% of individuals in certain sub-Saharan regions of Africa [25]. All affected individuals are homozygous for the *HbS* mutation of the hemoglobin gene. Despite the fact that the clinical expression of *SCD* can be influenced by environmental factors such as strenuous exercise, high-altitude, infection, and dehydration, *SCD* is fundamentally a genetic disorder.

Second, each year, 5–20% of the population in North America gets the flu [25]. Although the genetic make-up might make one person more or less susceptible to a particular year's variant, presumably, everyone could develop the flu if they had a sufficient exposure to the influenza virus. Therefore, despite the possible genetic differences in susceptibility, the flu is fundamentally an environmental (infectious) disease.

Third, the life-time probability of breast cancer in the US is ~12.5% in women and ~0.1% in men. Individuals (especially women) who carry the *BRCA1* or *BRCA2* mutations (<1% of the population) have 4–7 times the risk as that in the general population [25]. Nevertheless, presumably, there is a baseline risk of breast cancer such that no one is completely risk-free. Although the genetic make-up (including gender) influences the baseline risk and the environment likely affects the penetrance of the *BRCA* mutations, some breast cancer cases are fundamentally genetic and others are fundamentally environmental (of unclear type, but possibly due to exposures such as by toxins, radiation, pregnancy, or other occurrences).

Fourth, the human immunodeficiency virus (*HIV*) can infect anyone in the population although individuals who engage in certain high-risk behaviors (e.g., having unprotected anal-receptive sex or using intravenous drugs and sharing needles) are particularly susceptible. Among persons of northern European extraction, ~1% are homozygous for the *Δ-32 mutation* of the *CCR5* gene and are almost completely resistant to *HIV* [25]. Consequently, *HIV* infection is fundamentally an environmental disorder (infectious) with an interaction between two environmental factors (i.e., the virus and specific high-risk behaviors). However, certain genetic traits (*e.g., the Δ-32 mutation*) can be decisive in determining the degree of susceptibility.

Whether susceptibility to MS resembles any of these disease-states (or some other) is unknown although its polygenic nature is certain [5–14]. Nevertheless, several basic epidemiological observations in MS bear directly on the different possibilities. In this paper, we utilize directly observable, and well-established, "population parameters" (e.g., the concordance rates in twins and siblings, the proportion of women among MS patients, the population prevalence of MS, the time-dependent changes in the sex-ratio, etc.) to logically infer the values of other non-observable parameters of interest (e.g., the population probability of being genetically susceptible, the likelihood that a susceptible person actually develops MS, the proportion of

susceptible individuals who are women, the likelihood that a susceptible individual experiences a sufficient environmental exposure, etc.).

## Methods

For the purpose of this analysis we define, explicitly, five general terms (*Table 1*) and, in addition, provide a set of parameter abbreviations to be used for the purposes of notational simplicity (*Table 2*). The first term is {$P(MS)$}, which represents the expected life-time probability that a random individual from the general population ($Z$) will develop MS {i.e., the expected penetrance of MS is $P(MS) = P(MS|Z)$}. As discussed below, this parameter is related to the population prevalence.

 The second term is {$P(G)$}, which represents the expected probability that a random individual from ($Z$) is also a member of the ($G$) subset– i.e., $P(G) = P(G|Z)$. In turn, we define the ($G$) subset to include everyone who has any non-zero chance of developing MS (i.e., regardless of how small that risk might be). All individuals who are not in the subset ($G$) are considered to be in the mutually exclusive subset ($G-$) of non-susceptible individuals who have no chance of getting MS, regardless of their environmental experiences. We also define the set {$X$} to be the set of penetrance values for members of the ($G$) subset. If the variance of penetrance values

**Table 1. Definitions for epidemiological parameters used in the analysis.**

| Parameter | Definition |
|---|---|
| ($Z$) | Set of all individuals in the population |
| $P(MS)$ | Expected life-time probability of developing *MS* for a member of ($Z$) |
| ($G$) | Subset of all individuals in ($Z$) who have any non-zero chance of getting MS |
| ($G-$) | Subset of all individuals in ($Z$) who have no chance of getting MS |
| ($G1$) | The subset of "high-penetrance" individuals in ($G$) |
| ($G2$) | The subset of "low-penetrance" individuals in ($G$) such that: ($G1$) + ($G2$) = ($G$) |
| {$X$} | Set of all penetrance values for members of the subset ($G$) |
| ($H+$) | Set of all carriers of the $DRB1^*15:01~DQB1^*06:02~a1$ haplotype in ($Z$) |
| ($H-$) | Set of all non-carriers of the ($H+$) HLA haplotype in ($Z$) |
| ($M$) | Set of all men in ($Z$) |
| ($F$) | Set of all women in ($Z$) |
| $P(E)$ | Expected probability of an environmental exposure "sufficient to cause MS" in the subset ($G$), given the prevailing environmental conditions of the time |
| ($E_T$) | The prevailing environmental conditions of during a specific time-period ($T$) |
| ($E_1$) | That part of the sufficient environmental exposure shared exclusively by *MZ*- or *DZ*-twins– especially during the *IU* and early post-natal period |
| ($E_2$) | That part of the sufficient environmental exposure shared by the population generally: |
| ($E_3$) | The potential part of a sufficient environmental exposure due exclusively to the shared microenvironment of families. However, observationally [62–68]: $P(E) = P(E_1, E_2, E_3) = P(E_1, E_2)$ |
| ($E_i$) | The environmental exposure sufficient to cause MS in the $i^{th}$ individual in ($G$) |
| $P(MS\|MZ_{MS})$ | Expected life-time probability of developing *MS* for an *MZ*-twin whose co-twin either has, or will develop, *MS* |
| $P(MS\|DZ_{MS})$ | Expected life-time probability of developing *MS* for a *DZ*-twin whose co-twin either has, or will develop, *MS* |
| $P(MS\|S_{MS})$ | Expected life-time probability of developing *MS* for a sibling whose co-sibling either has, or will develop, *MS* |
| $P(MS\|IG_{MS})$ | Value of $P(MS\|MZ_{MS})$, which has been adjusted to exclude the impact of the similar *IU* and early post-natal environments of *MZ*-twins |
| $P(MZ_{MS})$ $P(IG_{MS})$ | Expected life-time probability of developing *MS* for an individual from any *MZ*-twin-ship. $P(MZ_{MS}) = P(IG_{MS}) = P(MS)$ |

**Table 2. Principal parameter abbreviations.**

| Parameter | Definition |
|---|---|
| $x_i$ | $\forall G_i \in G : P(MS|G_i) = x_i$– the penetrance of the ($i^{th}$) genotype in (G) |
| $\{X\}$ | Set of all penetrance values ($x_i$) in the (G) subset |
| $x$ | $P(MS|G)$ —the expected penetrance for the (G) subset |
| $x'$ | $P(MS|IG_{MS})$ —the expected penetrance for the ($IG_{MS}$) subset |
| $\sigma_X^2$ | Variance of the penetrance values for the (G) subset— $Var(X)$ |
| $x_1$ | $P(MS|G1)$ —the expected penetrance for the (G1) subset |
| $x_1'$ | $P(MS|G1, IG_{MS})$ —the expected penetrance for the (G1, $IG_{MS}$) subset |
| $\sigma_{x1}^2$ | Variance of the penetrance values for the (G1) subset |
| $x_2$ | $P(MS|G2)$ —the expected penetrance for the (G2) subset |
| $x_2'$ | $P(MS|G2, IG_{MS})$– the expected penetrance for the (G2, $IG_{MS}$) subset |
| $\sigma_{x2}^2$ | Variance of the penetrance values for the (G2) subset |
| $h(u)$ | Hazard function for men—where: $u = P(E)$ |
| $g(u)$ | Hazard function for women—where: $u = P(E)$ |
| $R$ | $= g(u)/h(u)$ —proportionality constant for hazard |
| $C$ | $P(MS)_1/P(MS)_2$ —ratio of $P(MS)$ at Timepoint-1 to that at Timepoint-2 |
| $p$ | $P(G1|G)$ —the proportion of the (G) subset that is also in (G1) |
| $a$ | $= (x_1/x)$ |
| $b$ | $= (x_2/x)$ |
| $v$ | $= (x_1'/x')$ |
| $w$ | $= (x_2'/x')$ |
| $r$ | $= (x_1'/x_1)$ |
| $s$ | $= (x_2'/x_2)$ |
| $t$ | $= P(G1|MS)/P(G2|MS)$ |
| $c$ | $= P(MS|G,E,M)$ - the limiting value of the exponential curve for men |
| $d$ | $= P(MS|G,E,F)$- the limiting value of the exponential curve for women |

in $\{X\}$ is non-zero then, for at least one partition, the subset (G) can be divided into two mutually exclusive sub-subsets, (G1) and (G2), suitably defined, such that the expected penetrance for the sub-subset (G1) is greater than that for (G2). If this difference in expected penetrance between the two sub-subsets is statistically significant, we will restrict our analysis to those circumstances, in which the sub-subsets {(G1) and (G2)}, considered separately, each has a penetrance distribution, which conforms to the Upper Solution (*see Proposition #1, below*). Although such an analysis focuses on possible unimodal and bimodal distributions for the set $\{X\}$, this constraint does not impose a bimodal distribution on it. Rather, the distribution of the set $\{X\}$ could still be unimodal, bimodal, trimodal or multi-modal {*NB: however, if the set $\{X\}$ is bimodal and the subset (G−) is non-empty, then the population (Z) has a trimodal distribution of penetrance values*}.

The third term is $\{P(E)\}$, which represents the expected probability that a member of the (G) subset will experience an environmental exposure sufficient to cause MS, given the prevailing environmental conditions of the time ($E_T$) – i.e., $P(E) = P(E|G,E_T)$. Using this definition for environmental exposure, even for those circumstances in which MS is either "purely genetic" or "purely environmental", we note that in all cases: $P(MS,E) = P(MS)$.

The fourth is a set of related terms $\{P(MS|MZ_{MS}), P(MS|DZ_{MS})$, and $P(MS|S_{MS})\}$. The 1st two terms, $\{P(MS|MZ_{MS})\}$ and $\{P(MS|DZ_{MS})\}$, represent the expected conditional lifetime probability of developing MS for an individual from either a monozygotic (MZ) or a dizygotic (DZ) twin-ship, given the fact that their co-twin either has or will develop MS. These

probabilities are estimated by the observed proband-wise concordance rate for either *MZ*-twins or *DZ*-twins [26]. In a similar manner, the term {$P(MS \mid S_{MS})$} represents the expected conditional life-time probability of developing MS in a sibling (*S*), given the fact that their co-sibling either has or will develop MS.

Last, is the term {$P(MS \mid IG_{MS})$}, which represents the adjusted proband-wise concordance rate for *MZ*-twins. Such an adjustment may be necessary because concordant *MZ*-twins, in addition to sharing their identical genotypes (*IG*), also share the intrauterine (*IU*) and certain other (especially early) post-natal environments. Thus, it is possible that these shared environmental experiences of *MZ*-twins might significantly impact the likelihood of their developing MS in the future. One method to estimate the adjustment necessary in such a circumstance is to consider the difference in concordance rates between non-twin siblings and fraternal twins (i.e., siblings who share the same genetic relationship but who are divergent in their *IU* and certain post-natal experiences). Although epidemiological studies have differed somewhat with regard to the magnitude of any such differences [27–34], population-based studies out of Canada suggest that the impact of these shared environmental events may be substantial [29]. As demonstrated in the *S1 File* (*#1*), we can use the observed recurrence-rate data to make this adjustment such that:

$$P(MS|IG_{MS}) = \{P(MS|S_{MS})/P(MS|DZ_{MS})\}*P(MS|MZ_{MS})$$

From these definitions and relationships, we can use well-established values for the different population parameters to logically deduce the value of the another, non-observable, parameter {$P(MS|G)$}, which represents the conditional life-time probability of developing MS for a member of the (*G*) subset. This term is referred to as the expected penetrance for the (*G*) subset. We note that, from the definition of the (*G*) subset, everyone who actually develops MS during their life-time must belong to this subset. Therefore, the joint probability {$P(MS, G)$} must be the same as {$P(MS)$}, so that, by definition:

$$P(MS, G) = P(MS) \text{ and, analogously}: \quad P(MZ_{MS}, G) = P(MZ_{MS})$$

From this, and from the definition of conditional probability:

$$P(MS|G) = P(MS, G)/P(G) = P(MS)/P(G)$$

This equation can be re-arranged to yield:

$$P(G) = P(MS)/P(MS|G)$$

This relationship, once established, can then be used to assess the nature of MS pathogenesis. For example, if {$P(G) = 1$}, then anyone can get the disease under the right environmental circumstances (*e.g.*, *flu, breast cancer, & HIV*) and we would conclude that MS must, in some cases, be caused by "purely environmental" factors. Notably, however, such circumstance does not preclude the possibility that genetic factors strongly influence the likelihood of disease (*e.g.*, *breast cancer & HIV*).

By contrast, if {$P(G)<1$}, this indicates that only certain individuals can possibly get the disease (*e.g.*, *SCD*) and, therefore, that MS must be a genetic disorder (i.e., unless a person has the correct genetic make-up, they have no chance, whatsoever, of getting the disease, regardless of their environmental exposure). Naturally, also, such a conclusion would have no bearing on whether disease pathogenesis also requires the co-occurrence of specific environmental events. Also, in this circumstance, how we might characterize the nature of genetic susceptibility, would depend upon the degree to which *P(G)* was less the unity and upon the magnitude of the disparity between any so-called "high" and "low" penetrance subgroups. For example, in

*HIV*, if homozygous *Δ-32* mutations (occurring in 1% of a northern European population) were completely protective, then: $P(G) = 0.99$. In this circumstance, however, we would likely characterize the disease as being fundamentally environmental and the homozygous *Δ-32* mutations as being protective rather characterizing every other genotype as being "suscepti-ble". By contrast, in *SCD*, where: $P(G) = 0.03$ –i.e., 3% of certain African populations–we would characterize carrying homozygous *HbS* mutations as the defining trait for membership in the "genetically-susceptible" subset (*G*). Even if it were possible, in extremely rare circum-stances, for an individual to develop *SCD* in the absence of homozygous *HbS* mutations, we would still consider this disease to be fundamentally genetic.

## Results

### 1. MS penetrance in the general population–*P(MS)*

***Conclusion***: $P(MS) \approx 0.003$

   ***Argument***: One possible estimate of *P(MS)* could be the prevalence of MS in a population. However, because the clinical onset of MS occurs largely between the ages of 15 and 45 years (e.g., *Fig 1*), the measured cross-sectional prevalence of MS (using the entire population as the denominator) will necessarily include individuals with different likelihoods of having already

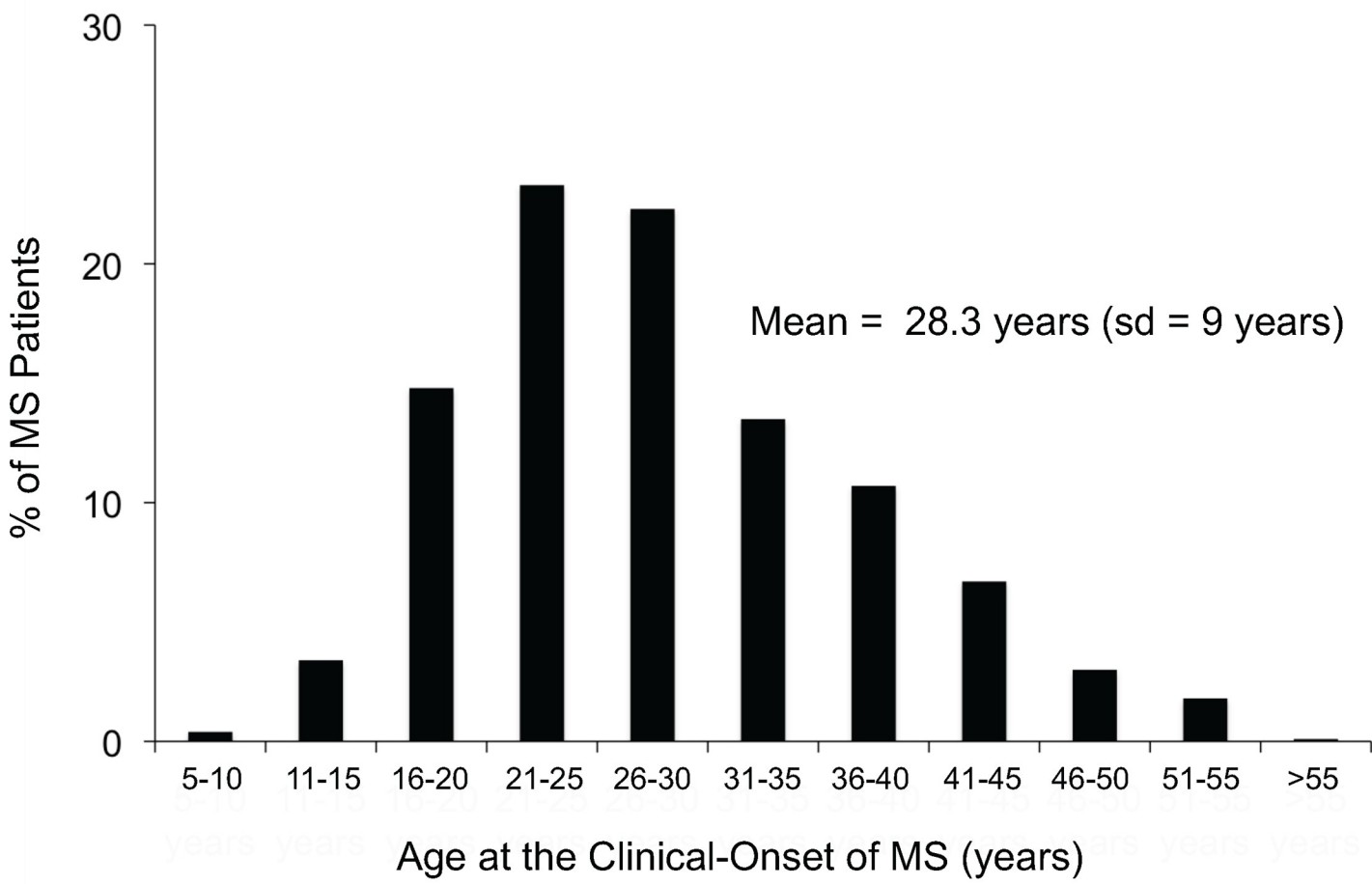

**Fig 1. The distribution of the age at the clinical-onset of disease in a cohort of 1,463 patients with MS (sd = standard deviation).** *Data from Liguori et al.* [35].

[In thousands, except as indicated (226,546 represents 226,546,000). As of April 1. Excludes Armed Forces overseas. For definition of median, see Guide to Tabular Presentation]

| Age | 1980[1] | | | 1990[2] | | | 2000[3] | | | 2010 | | |
|---|---|---|---|---|---|---|---|---|---|---|---|---|
| | Total | Male | Female | Total | Male | Female | Total | Male | Female | Total | Male | Female |
| Total. . . . . . . . . . . | 226,546 | 110,053 | 116,493 | 248,791 | 121,284 | 127,507 | 281,425 | 138,056 | 143,368 | 308,746 | 151,781 | 156,964 |
| Under 5 years . . . . | 16,348 | 8,362 | 7,986 | 18,765 | 9,603 | 9,162 | 19,176 | 9,811 | 9,365 | 20,201 | 10,319 | 9,882 |
| 5 to 9 years . . . . . . | 16,700 | 8,539 | 8,161 | 18,042 | 9,236 | 8,806 | 20,550 | 10,523 | 10,026 | 20,349 | 10,390 | 9,959 |
| 10 to 14 years . . . . | 18,242 | 9,316 | 8,926 | 17,067 | 8,742 | 8,325 | 20,528 | 10,520 | 10,008 | 20,677 | 10,580 | 10,097 |
| 15 to 19 years . . . . | 21,168 | 10,755 | 10,413 | 17,893 | 9,178 | 8,714 | 20,219 | 10,391 | 9,828 | 22,040 | 11,304 | 10,737 |
| 20 to 24 years . . . . | 21,319 | 10,663 | 10,655 | 19,143 | 9,749 | 9,394 | 18,963 | 9,688 | 9,275 | 21,586 | 11,014 | 10,572 |
| 25 to 29 years . . . . | 19,521 | 9,705 | 9,816 | 21,336 | 10,708 | 10,629 | 19,382 | 9,799 | 9,583 | 21,102 | 10,636 | 10,466 |
| 30 to 34 years . . . . | 17.5decreased | | | 21,838 | 10,866 | 10,973 | 20,511 | 10,322 | 10,189 | 19,962 | 9,997 | 9,966 |
| 35 to 39 years . . . . | 13,965 | 6,862 | 7,104 | 19,851 | 9,837 | 10,014 | 22,707 | 11,319 | 11,388 | 20,180 | 10,042 | 10,138 |
| 40 to 44 years . . . . | 11,669 | 5,708 | 5,961 | 17,593 | 8,679 | 8,914 | 22,442 | 11,130 | 11,313 | 20,891 | 10,394 | 10,497 |
| 45 to 49 years . . . . | 11,090 | 5,388 | 5,702 | 13,747 | 6,741 | 7,006 | 20,093 | 9,890 | 10,203 | 22,709 | 11,209 | 11,500 |
| 50 to 54 years . . . . | 11,710 | 5,621 | 6,089 | 11,315 | 5,494 | 5,821 | 17,586 | 8,608 | 8,978 | 22,298 | 10,933 | 11,365 |
| 55 to 59 years . . . . | 11,615 | 5,482 | 6,133 | 10,489 | 5,009 | 5,480 | 13,469 | 6,509 | 6,961 | 19,665 | 9,524 | 10,141 |
| 60 to 64 years . . . . | 10,088 | 4,670 | 5,418 | 10,627 | 4,947 | 5,679 | 10,806 | 5,137 | 5,669 | 16,818 | 8,078 | 8,740 |
| 65 to 74 years . . . . | 15,581 | 6,757 | 8,824 | 18,048 | 7,908 | 10,140 | 18,391 | 8,303 | 10,088 | 21,713 | 10,097 | 11,617 |
| 75 to 84 years . . . . | 7,729 | 2,867 | 4,862 | 10,014 | 3,745 | 6,268 | 12,361 | 4,879 | 7,482 | 13,061 | 5,477 | 7,584 |
| 85 years and over . . . . . . . . . . | 2,240 | 682 | 1,559 | 3,022 | 841 | 2,181 | 4,240 | 1,227 | 3,013 | 5,493 | 1,790 | 3,704 |
| 5 to 13 years . . . . . . | 31,159 | 15,923 | 15,237 | 31,839 | 16,301 | 15,538 | 37,026 | 18,964 | 18,062 | 36,860 | 18,834 | 18,026 |
| 14 to 17 years . . . . . | 16,247 | 8,298 | 7,950 | 13,345 | 6,860 | 6,485 | 16,093 | 8,285 | 7,808 | 17,120 | 8,792 | 8,328 |
| 18 to 24 years . . . . | 30,022 | 15,054 | 14,969 | 26,961 | 13,744 | 13,217 | 27,141 | 13,873 | 13,268 | 30,672 | 15,662 | 15,010 |
| 18 years and over . . | 162,791 | 77,473 | 85,321 | 184,841 | 88,519 | 96,322 | 209,130 | 100,996 | 108,133 | 234,564 | 113,836 | 120,728 |
| 55 years and over . . | 47,253 | 20,458 | 26,796 | 52,200 | 22,450 | 29,748 | 59,267 | 26,055 | 33,212 | 76,751 | 34,964 | 41,787 |
| 65 years and over . . | 25,550 | 10,306 | 15,245 | 31,084 | 12,494 | 18,589 | 34,992 | 14,410 | 20,582 | 40,268 | 17,363 | 22,905 |
| 75 years and over . . | 9,969 | 3,549 | 6,421 | 13,036 | 4,586 | 8,449 | 16,601 | 6,106 | 10,495 | 18,555 | 7,266 | 11,288 |
| Median age (years) . . . . . . . . . | 30.0 | 28.8 | 31.3 | 32.8 | 31.6 | 34.0 | 35.3 | 34.0 | 36.5 | 37.2 | 35.8 | 38.5 |

[1] Total population count has been revised since the 1980 census publications. Numbers by age and sex have not been corrected. [2] The data shown have been modified from the official 1990 census counts. See text, this section, for explanation.
The April 1, 1990, estimates base (248,790,925) includes count question resolution corrections processed through August 1997. It generally does not include adjustments for census coverage errors. However, it includes adjustments estimated for the 1995 Test Census in various localities in California, New Jersey, and Louisiana, and the 1998 census dress rehearsals in localities in
California and Wisconsin. These adjustments amounted to a total of 81,052 persons. [3] The April 1, 2000 population estimates base reflects changes to the Census 2000 population from the Count Question Resolution program.
Source: U.S. Census Bureau, Current Population Reports, P25-1095; "Table US-EST90INT-04—Intercensal Estimates of the United States Resident Population by Age Groups and Sex, 1990–2000: Selected Months," September 2002, <http://www.census .gov/popest/archives/EST90INTERCENSAL/US-EST90INT-04.html>; and 2010 Census Redistricting Data (P.L. 94-171) Summary File, <http://www.census.gov/rdo/data/2010_census_redistricting_data_pl_94-171_summary_files.html>.

**Fig 2. US census data (for each decade from 1980 to 2010) for resident population by age and sex.** Age-band (*A1*) is highlighted in turquoise; Age-band (*A2*) is highlighted in yellow; and Age-band (*A3*) is highlighted in grey.

developed MS [35]. For example, using the 2010 United States census data (for the total resident population–*see Fig 2*) as an approximation, we can divide the general population (*Z*) into the three mutually exclusive age-bands (*A1*, *A2*, and *A3*), such that:

$$A1 = \{< 15 years\}; \qquad P(A1|Z) \approx 0.20$$
$$A2 = \{15 - 45 years\}; \quad P(A2|Z) \approx 0.41$$
$$A3 = \{> 45 years\}; \qquad P(A3|Z) \approx 0.39$$

Because so few of MS patients have their disease onset prior to the age of 15 years (e.g., *Fig 1*) it seems a reasonable approximation that:

$$P(MS|A1) \approx 0*P(MS)$$

By contrast, as noted above, the age group (15–45 years) accounts for the large majority of clinical onsets, which have a roughly symmetrical distribution with a mean of 28.3 years (*Fig 1*). If the distribution were exactly symmetrical and centered on 30 years, the measured prevalence in this age band would be ~50% of the value of *P(MS)*. Therefore, it seems reasonable to estimate:

$$P(MS|A2) \approx 0.5*P(MS)$$

For the older age band (>45 years) most patients will have already developed the disease (*Fig 1*). Thus, on the one hand, one might expect that the measured prevalence in this age-

band to be equal to *P*(*MS*). On the other hand, there is a small but definite excessive mortality in MS such that life expectancy is reduced in MS-patients by about 5–10 years [36–40] although a recent study from Denmark [41] reported that short-term survival has steadily improved for patients beginning in 1950 and continuing through 1999. This excessive mortality will make the estimate too small by some amount although it seems unlikely that this reduction will be more than 25%. Thus, a range of plausible estimates is likely to be:

$$0.75*P(MS) < P*(MS|A3) < P(MS)$$

Combining these three different estimates yields the estimate:

$$P(MS) \approx 0.20*P(MS|A1) + 0.41*P(MS|A2) + 0.39*P(MS|A3)$$

Defining the measured prevalence in the population as (*prev*), this estimate translates to:

$$1.7*(prev) < P(MS) < 2*(prev)$$

A second method to estimate *P*(*MS*) would be to use a measured prevalence for MS, which is restricted to the age-band of 45–54 years. Thus, within this age-band, almost all patients will have already experienced their clinical-onset and only a few will have experienced their (expected) excessive mortality. Consequently, by this method:

$$P(MS) \approx prev(age = 45 - 54\ years)$$

A third method would be to use population-based death data and to consider the percentage of death certificates that mention the diagnosis of MS (not necessarily as, but including, the immediate, underlying, or contributing cause). Thus, by the time of death, any case of clinically evident MS must, by definition, have already declared itself. Consequently, by this method:

$$P(MS) \approx \%of\ death\ certificates\ mentioning\ MS$$

In 2001, we took a cursory (*unpublished*) look at the Kaiser northern California database. At the time, there were 4,352 unique persons in the database with a diagnostic code for MS. With 2.9 million persons enrolled in Kaiser northern California at the time, and if this population is a representative sample, this would translate to an MS-prevalence in northern California of 150 per 100,000 population. Such an estimate is consistent with many other published studies in northern populations, which generally find the prevalence of MS to be 100–200 per 100,000 population [42]. A recent study from the United States, using multiple administrative health claims (AHC) datasets [43] estimated that the prevalence of MS in adults (i.e., age $\geq$ 18 years)–which represents ~75% of the US population (*see Fig 2*)–as 288–309 per 100,000 population. For comparison purposes, this translates to a prevalence in the entire population of 216–232 per 100,000 individuals.

Similarly, in a Swedish study by Sundström and co-workers [44], the age-specific prevalence of MS in the 45–54 year age-band was reported to be 304 per 100,000 population. In the AHC study [43], the estimate for this same age-band (considering the entire population) was 314–337 per 100,000 individuals.

And, finally, in a recent population-based multiple-cause-death study from British Columbia [45], a diagnosis of MS was mentioned on 0.28% of the death certificates.

Thus, all three of these methods of estimation are quite consistent with each other. The range of values supported, collectively, by these observations is:

$$0.0025 \leq P(MS) \leq 0.0046$$

The best support is for the conclusion that, in the northern populations of Europe and the Americas:

$$P(MS) \approx 0.003$$

However, despite the notable consistency of these three estimates, each of these methods relates only to "diagnosed" MS in the general population (*Z*). If undiagnosed (i.e., pathological) MS is included in the calculation [46–49], this estimate may increase by as much as 50–100% (*see #8 below*).

## 2. Adjusting for the shared environment of *MZ*-twins–$P(MS \mid IG_{MS})$

***Conclusion:*** $P(MS|IG_{MS}) = P(MS|G,IG_{MS}) \approx 0.134$

*Argument*: Most epidemiological studies in northern populations report the proband-wise concordance rate for *MZ*-twins to be in the range of 25–30% [27–34, 50]. Using the population-based data from Canada (*Table 3*; *Fig 3*), leads to the estimate of:

$$P(MS|MZ_{MS}) = P(MS, G|MZ_{MS}) = P(MS|G, MZ_{MS}) \approx 0.25$$

Suppose that each of the (*n*) individuals (*k* = 1,2,. . .,*n*) within the general population (*Z*), has a unique genotype ($G_k$), in which case:

$$P(MS|MZ_{MS}) = \sum_{k=1}^{n} P(MS, G_k|MZ_{MS}) = \sum_{k=1}^{n} P(G_k|MZ_{MS}) * P(MS|G_k, MZ_{MS})$$

We can then define the term ($IG_{MS}$) such that:

$$\forall G_k \in Z : \ P(MS|G_k, IG_{MS}) = P(MS|G_k)$$

$$\text{where} : \quad \forall G_k \in Z : \ P(G_k|IG_{MS}) = P(G_k|MZ_{MS})$$

**Table 3. Canadian population data on *MZ*-twin concordance broken down by (*H*+)-haplotype and gender-status [*].**

| MZ-Twins | | | |
|---|---|---|---|
| *HLA-DRB1*15 Status* | *H+* | *H–* | *Totals* |
| Concordant for MS (C) | 9 | 11 | 20 |
| Discordant for MS (D) | 31 | 42 | 73 |
| Totals | 40 | 53 | 93 |
| Pair-wise Concordance | 0.23 | 0.21 | 0.22 |
| Proband-wise Concordance | 0.31 | 0.29 | 0.30 |
| *Gender Status* | *Women* | *Men* | *Totals* |
| Concordant for MS (C) | 22 | 2 | 24 |
| Discordant for MS (D) | 66 | 43 | 109 |
| Totals | 88 | 45 | 133 |
| Pair-wise Concordance | 0.25 | 0.04 | 0.18 |
| Proband-wise Concordance | 0.34 | 0.067 | 0.25 |

[*] Data from Willer et al. [29] and Orton et al. [56]. The *MZ*-twins were drawn from the 19,938 MS-patients in the *CCGPSMS* database. The pair-wise concordance were calculated as: C/(C+D). The proband-wise concordance were calculated as: 2C/(2C+D); adjusted [26] for double ascertainments (13/24 = 54%).

## Summary of MS Data from Canada

### Population Data

$P(H+) = 0.24$

$P(F) = P(M) = 0.5$

### Family Data

$P(MS \mid MZ_{MS}) = 0.253$

$P(MS \mid DZ_{MS}) = 0.054$

$P(MS \mid S_{MS}) = 20/692 = 0.029$

### DRB1*15 Data

$P(H+ \mid MS) = P(H+ \mid MZ_{MS}) = 40/93 = 0.43$

$P(H+ \mid MS, MZ_{MS}) = 9/20 = 0.45$

$P(H+ \mid MS)/P(H+) = 0.43/0.24 = 1.79$

$P(H+ \mid MS, MZ_{MS})/P(H+ \mid MS) = 0.45/0.43 = 1.05$

$P(MS \mid H+, MZ_{MS}) = 0.31$

$P(MS \mid H-, MZ_{MS}) = 0.29$

### Gender Data

$P(F \mid MS) = P(F \mid MZ_{MS}) = 88/133 = 0.66$

$P(F \mid MS, MZ_{MS}) = 22/24 = 0.92$

$P(F \mid MS)/P(F) = 0.66/0.5 = 1.32$

$P(F \mid MZ_{MS})/P(F \mid MS) = 0.92/0.66 = 1.39$

$P(MS \mid F, MZ_{MS}) = 0.34$

$P(MS \mid M, MZ_{MS}) = 0.067$

### Sex Ratio Data

*Time Period* (#1) -- 1941–1945:  $P(F \mid MS)_1/P(M \mid MS)_1 = 2.2$

*Time Period* (#2) -- 1976–1980:  $P(F \mid MS)_2/P(M \mid MS)_2 = 3.2$

**Fig 3. Summary of epidemiological data regarding MS in Canada** *circa* **2000–2010.** The value for *P(H+)* was provided by D. Sadovnick, was based on 400 Canadian controls, and the rate was confirmed in a large transplant database (*personal communication*). The *F:M* sex-ratio in the general population of Canada was taken from the 2010 Canadian census. Recurrence risks for monozygotic (*MZ*) twins, dizygotic (*DZ*) twins, and siblings (*S*) were taken from the study of Willer et al. [29]. The other summary data was taken from *Table 3*, and/or from the study of Willer et al. [29]. The *F:M* sex-ratio among Canadian MS patients at each of the 5-year time-periods (1941–1945 & 1976–1980) was taken from the study of Orton et al. [56].

Therefore :  $P(MS|IG_{MS}) = \sum_{k=1}^{n} P(G_k|IG_{MS}) * P(MS|G_k, IG_{MS}) = \sum_{k=1}^{n} P(G_k|MZ_{MS}) * P(MS|G_k)$

This is just the expected "adjusted" penetrance for the ($MZ_{MS}$) subset. As discussed earlier and, as developed in the *S1 File* (#1), {$P(MS|IG_{MS})$} can be estimated from the difference in observed concordance rates between siblings and fraternal twins. Using the Canadian population-based data (*Table 3*; *Fig 3*) on the recurrence risks in non-twin siblings and *DZ*-twins

(concordance rates for siblings = 2.9%; concordance rates for $DZ$-twins = 5.4%) to make this adjustment (*see above*) leads to the estimate of:

$$P(MS|IG_{MS}) = (2.9/5.4)*0.25 = 0.25/1.86 = 0.134$$

## 3. Proposition #1

***Assertions***: 1. Upper Solution: $x'/2 < x \leq x'$; and Lower Solution: $0 < x < x'/2$

2. The variance of $\{X\}$ conforms to the conditions:

$$0 \leq \sigma_X^2 < (x'/2)^2 \text{ and} : \quad \sigma_X^2 = x(x' - x)$$

3. If the individual values for penetrance within the $(G)$ subset are distributed in a unimodal manner, then:

$$P(MS|IG_{MS})/1.15 \leq P(MS|G) \leq P(MS|IG_{MS}) = x'$$

4. If the penetrance values in $(G)$ are distributed in a non-unimodal manner, and if: $P(MS|G) \geq x'/2$, then the Upper Solution limits applies and:

$$\forall \{x > x'/2\} : \; p < (2 - b^2 s)/(a^2 r - b^2 s)$$

5. And finally, for more extreme non-unimodal distributions of $(G)$–i.e., $P(MS|G) < x'/2$– then the Lower Solution applies and:

$$\forall \{x < x'/2\} : \; p > (2 - b^2 s)/(a^2 r - b^2 s)$$

***Proof***: For notational simplicity, as discussed previously, we use abbreviated terms for several parameters (*see Table 2*). Among the $(n)$ individuals in the general population $(Z)$, we have already defined the $(G)$ subset, which consists of everyone who has any non-zero lifetime probability of developing MS. Thus, each of the $(m)$ individuals in the $(G)$ subset $(i = 1,2,\ldots,m)$ has a unique genotype $(G_i)$, such that:

$$\forall G_i \in G : \; P(MS|G_i) > 0 \; \text{and, thus} : \; P(MS|G) > 0$$

We define (*see Table 2*) the parameters, $(x_i)$ and $(x)$, such that:

$$\forall G_i \in G : \; P(MS|G_i) = x_i \; \text{and} : \; P(MS|G) = x$$

Thus, $(x_i)$ represents the expected penetrance for MS in the $i^{th}$ individual of the $(G)$ subset. Even if this penetrance exactly matches that of another person, $(x_i)$ is still unique to the $i^{th}$ individual. Also, considering the penetrance values for each of the members of the $(G)$ subset, we can define the set $\{X\}$ such that:

$$X = \{x_i\} \; \text{and} : \; E(x_i) = E(X) = P(MS|G) = x$$

Because the $(G)$ subset forms a partition of the population $(Z)$, each of the $(m_2 = n - m)$ individuals, who are not in the $(G)$ subset, belongs to the mutually exclusive "non-susceptible" subset $(G-)$. Moreover, each of the $(m_2)$ individuals in the $(G-)$ subset $(j = 1,2,\ldots,m_2)$ has a unique genotype $(G_j)$, which has a zero conditional life-time probability of developing MS, so that:

$$\forall G_j \in G- : \; P(MS|G_j) = 0 \; \text{and, thus} : \; P(MS|G-) = 0$$

Also, if $\{Var(X) \neq 0\}$, we can partition the subset $(G)$ into two mutually exclusive sub-subsets, $(G1)$ and $(G2)$, suitably defined, such that the sub-subset $(G1)$ has a penetrance greater

than that of ($G2$). Again, for ease of notation, we define the quantities ($x'$,$x_1$,$x_1'$,$x_2$, & $x_2'$)– *see Table 2* –such that:

$$x' = P(MS|IG_{MS}) = P(MS|G, IG_{MS}) = P(MS, G|G, IG_{MS})$$

$$x_1 = P(MS|G1); \quad x_1' = P(MS|G1, IG_{MS})$$

$$\text{and}: \quad x_2 = P(MS|G2); \quad x_2' = P(MS|G2, IG_{MS})$$

In earlier iterations of this analysis [3, 4, 51, 52], we defined the subset ($G$) differently–i.e., $\forall G_i \in G$: $P(MS|G_i) \geq P(MS)$. We have chosen the current definition because it considerably simplifies the biological interpretation of the findings. Nevertheless, we note that, when circumstances fit the conditions of the <u>Lower Solution</u> (*see below*), the new sub-subset ($G1$) is, effectively, identical to the subset ($G$) defined earlier.

We define the term {$P(MZMS)$} to represent the life-time probability of developing MS for any single individual from an *MZ* twin-ship (i.e., where the status of their co-twin is unknown). Because identical twinning is considered non-hereditary [53], we expect that:

$$P(MZ_{MS}) = P(IG_{MS}) = P(MS)$$

As noted earlier, we also define the set {$X$} set to consist of the individual *MS*-penetrance values for all members of the ($G$) subset. Thus, the variance ($\sigma_X^2$) of the set {$X$} can be expressed as:

$$\sigma_X^2 = Var(X) = E(x_i - x)^2 = E(x_i)^2 - x^2$$

$$\text{Also}: \quad \forall G_i \in G: \ P(G_i|G) = 1/m; \ \ P(G) = m/n; \ \ E(x_i) = \sum_{i=1}^{m}(x_i)*(1/m)$$

$$\text{and}: \ \ E(x_i^2) = \sum_{i=1}^{m}(x_i^2)*(1/m) = x^2 + \sigma_X^2$$

It follows directly from the definitions of {$P(G)$} and {$P(MS|IG_{MS})$}–*see Methods & #2, above*–that:

$$P(MS|G_i, IG_{MS}) = P(MS|G_i, G, IG_{MS}) = P(MS|G_i, G) = P(MS|G_i) = x_i$$

1. Therefore, the probability {$P(MS,G_i|G,IG_{MS})$} can be re-written as:

$$P(MS, G_i|G, IG_{MS}) = P(G_i|G, IG_{MS})*P(MS|G_i, G, IG_{MS})$$
$$= P(G_i|G, IG_{MS})*P(MS|G_i) = P(G_i|G, IG_{MS})*(x_i)$$

2. In turn, the term $P(G_i|G,IG_{MS})$ can be re-written as:

$$P(G_i|G, IG_{MS}) = P(G_i|G, MS) = P(G_i, G, MS)/P(MS, G)$$
$$= P(MS|G_i, G)*P(G_i, G)/P(MS, G)$$
$$= (x_i)*P(G_i|G)/P(MS|G) = (x_i)*(1/m)/x$$

Combining these two Equations (*i.e.,* *1 & 2 above*) yields:

$$P(MS, G_i | G, IG_{MS}) = \{(x_i) * (1/m)/x\} * (x_i) = (x_i)^2 * (1/m)/x$$

However :
$$x' = P(MS|G, IG_{MS}) = P(MS, G|G, IG_{MS}) = \sum_{i=1}^{m} P(MS, G_i | G, IG_{MS})$$

Where :
$$\sum_{i=1}^{m} P(MS, G_i | G, IG_{MS}) = \sum_{i=1}^{m} (x_i^2) * (1/m)/x = E(x_i^2)/x$$

Consequently :
$$x' = (x^2 + \sigma_X^2)/x = x + \sigma_X^2/x$$

and, with rearrangement :
$$\sigma_X^2 = x(x' - x)$$

Notably, this equation can also be rearranged to yield a quadratic in $(x)$ of:

$$x^2 - (x')x + \sigma_X^2 = 0$$

In turn, this quadratic equation can be solved to yield: $x = (x'/2) \pm (\sqrt{(x')^2 - 4\sigma_X^2})/2$
which has real, non-negative, solutions only for: $0 \leq \sigma_X^2 \leq (x')^2/4 = (x'/2)^2$
The maximum variance for any distribution [54, 55] on the closed interval $[a,b]$ is:

$$\sigma^2 \leq (b - a)^2/4$$

Consequently, the maximum variance for the set $\{X\}$ is identical to that for the interval $[0,x']$, which is:

$$\sigma^2 = (x' - 0)^2/4 = (x'/2)^2$$

In addition, this maximum variance, $(x'/2)^2$, occurs when the distribution of penetrance values in the set $\{X\}$ is bimodal [54, 55], such that half the $(G)$ subset has a penetrance of $(0)$ and the other half has a penetrance of $(x')$. From this point of maximum variance, the variance of the $\{X\}$ subset decreases both when:
$x \rightarrow x'$ and: $x > x'/2$ (*the Upper Solution*)
and when: $x \rightarrow 0$ and: $x < x'/2$ (*the Lower Solution*)
By definition, any solution requiring $\{P(MS|G_i) = 0\}$ for any portion of $(G)$ is excluded.
Therefore, the Upper Solution limits become: $x'/2 < x \leq x'$
And the Lower Solution limits become: $0 < x < x'/2$
Moreover because: $x' = x + \sigma_X^2/x$ Therefore, if: $\sigma_X^2 = 0$; then: $x' = x$
Using abreviated notations (*Table 2*), notably, there are three other related equivilences:

$$P(MS) = P(G1)x_1 + P(G2)x_2 \tag{Eq 1A}$$

$$x = P(G1|G)x_1 + P(G2|G)x_2 = px_1 + (1 - p)x_2 \tag{Eq 1B}$$

and : $$x' = P(G1|IG_{MS})x_1' + P(G2|IG_{MS})x_2' = pax_1' + (1 - p)bx_2' \tag{Eq 1C}$$

Because, by definition ($x_1 > x_2$) –see *Methods* –therefore, applying *Eq 1B*:

$$x_1 > x > x_2$$

also: if: $x_2 > x'/2$; then $x > x'/2$; and the distribution of $\{X\}$ will conform to the Upper Solution (*see above*).

Also, applying *Eq 1C*: if: ($x_1' > x_2'$); then: ($x_1' > x' > x_2'$)

**The Upper Solution.** The Upper Solution, as: ($x \to x'$), represents the gradual transition from a bimodal distribution to a unimodal distribution and, ultimately, to a distribution, in which every genotype in ($G$) has exactly the same penetrance (i.e., $x = x'$). As noted earlier (*above*), the Upper Solution requires that:

$$P(MS|G) = x > x'/2 = P(MS|IG_{MS})/2$$

Alternatively, we can define ($p, a, b, r, \& s$) –see *Table 2* –such that:

$$p = P(G1|G); \; a = x_1/x; \; b = x_2/x; \; r = x_1'/x_1; \; \text{and} : \; s = x_2'/x_2$$

and, as shown in the *in S1 File (#3b)*, the Upper Solution applies whenever:

$$\forall \{x > x'/2\} : \; p < (2 - b^2 s)/(a^2 r - b^2 s)$$

Also, as demonstrated by others [54], the maximum variance of <u>any</u> unimodal distribution on the closed interval $[a,b]$ is: $\sigma^2 \leq (b-a)^2/9$. Considering a unimodal distribution on the interval $(0,x']$, therefore: $\sigma_X^2 < (x')^2/9$

Substituting this limit into the upper quadratic solution (*above*)–assuming this limit applies equally to the set $\{X\}$–yields:

$$x \geq (x'/2) + (\sqrt{(x')^2 - 4*(x')^2/9})/2 = (0.50 + 0.37)*x' = 0.87*x'$$

Consequently, in order for $\{X\}$ to have a unimodal distribution requires that:

$$0.87*x' = x'/1.15 \leq x = P(MS|G) \leq x' = P(MS|IG_{MS})$$

**The Lower Solution.** By contrast, the Lower Solution as: ($x \to 0$), represents an increasingly assymetric non-unimodal distribution of penetrance values within the ($G$) subset. Nevertheless, as noted *above*, all Lower Solutions require that:

$$P(M|G) = x < x'/2 = P(MS|IG_{MS})/2$$

Alternatively (*as above*), using the parameters ($p, a, b, r, \& s$) –see *Table 2* –the Lower Solution applies whenever:

$$\forall \{x < x'/2\} : \; p > (2 - b^2 s)/(a^2 r - b^2 s)$$

Because, by definition, ($x_1 > x_2$), and because, we assume that sub-subsets ($G1$) and ($G2$) having different penetrances, considered separately, conform to an Upper Solution (*see Methods*) therefore:

$$x_1'/2 < x_1 \leq x_1' \text{ and} : \; x_2'/2 < x_2 \leq x_2'$$

and, consequently: $x_1' < 2x_1$ and: $x_2' < 2x_2$.

In this case, the difference ($x_2' - x_1' \geq 0$) will be at its maximum when ($x_1$) has its minimum variance ($x_1' = x_1$) and ($x_2$) has its maximum variance ($x_2' \approx 2x_2$). Moreover, this difference will be (0) at the point where ($x_2$) is slightly more than half of ($x_1'$). At this point: ($2x_2 > x_1' = x_2'$),

and application of *Eq 1C* (*above*) yields:

$$x' < P(G1|IG_{MS})2x_2 + P(G2|IG_{MS})2x_2 = 2x_2$$

or: $x_2 > x'/2$

As magnitude of the difference $(x_2'-x_1')$ increases, considering the same variances for $(x_1)$ and $(x_2)$, the lower limit of $(x_2)$ will increase relative to $(x_2')$. Similarly, as the variance of $(x_2)$ decreases, the lower limit of $(x_2)$ will increase relative to $(x_2')$ and the the value of $(x_2)$ at point where $(x_2'-x_1' = 0)$ will be greater. Consequently: $\forall(x_2'-x_1')\geq0$: $x_2 > x'/2$ and the Upper Solution applies.

Therefore, for all Lower Solutions: $x_1' > x_2'$ and from *Eq 1C* (*above*): $x_1' > x'$.

Notably, the values of $\{x', x_1', x_2', \& P(MS)\}$ represent observed population parameters (or are drrived from observed parameters). As such, these values shoud be considered as "fixed" although, naturally, there is always the possiblity of error in their observation.

**Breast cancer.** As an example, it is instructive to apply this same analysis to the risk in women of developing breast cancer (*descsribed briefly in the Introduction*). Clearly, this distribution is bimodal with <1% of women possessing the *BRCA* mutations, and with these individuals having 4–7 times the risk of breast cancer as that for everyone else. For this analysis, we assume that the subsets of women with ($G1$) and without ($G2$) *BRCA* mutations have a uniform penetrance within each subset. Also, we will also use parameter values that conform to the known epidemiology of breast cancer in women (*BC*) such that:

$$P(BC) = 0.125; \quad P(BC|G1) = 0.7 \text{ and} : \quad P(G1) = 0.01$$

Under these conditions, and in all circumstances, it is the case that:

$$0.18 \leq P(G) \leq 1; \quad 0.15 \leq x' \leq 0.7 \text{ and} : \quad 0.83*x' \leq x \leq x'$$

Although, unlike MS, we don't have "observational" estimates for adjusted the *MZ*-twin recurrence risk ($x'$), these circumstances for breast cancer, clearly, conform to the upper solution of the quadratic equation (*above*). For example, if this recurrence risk were (~15%) then: $\{P(G) = 1\}$ and: $\{x = 0.83*x'\}$. In this case, the fact that the distribution is bimodal is confirmed by the fact that the value of ($x$) is below the lower limit for a unimodal distribution (*see above*). By contrast, if all breast cancers are, to some degree, genetic disorders–{i.e., if: $(P(G)<1)$}– then, as $P(G)$ decreases, the value of ($x$) will increase. Nevertheless, the bimodality of the distribution will still be evident down to $P(G) = 0.86$. Below this point, however, the bimodal nature of the distribution will no longer be distinguishable (purely by consideration of the variance) from a unimodal distribution. Regardless, however, using these parameter values, the distribution would not actually become unimodal until the point at which: $\{x = x'\}$.

## 4. Genetic susceptibility to MS–general considerations

**4a. The Upper Solution.** *Conclusions*: $0.022 \leq P(G) \leq 0.045$

*Argument*: From the Upper Solution in *Proposition #1* and in conjunction with our estimate from *#2* (*above*) for $\{P(MS|IG_{MS})\}$, it follows directly, that:

$$0.134/2 = 0.067 < x = P(MS|G) \leq 0.134$$

We can then apply the relationship developed in the *Methods* that:

$$P(G) = P(MS)/P(MS|G)$$

With this we have all the data necessary to establish the limits for the percentage of the population who are members of the ($G$) subset. Thus, using this range for $P(MS|G)$, together with

**Table 4. MS prevalence, *MZ*-twin concordance {$P(MS \mid MZ_{MS})$}, and "genetic susceptibility" {$P(G)$}–for the upper solution (*see Text*)–in different geographical locations.**

| Geographical Location | MS Prevalence* | $P(MS \mid MZ_{MS})$[†] | Latitude | $P(G)$[††] |
|---|---|---|---|---|
| *North America* | | | | |
| Canada [29] | 68–248 | 0.253 | N45-60˚ | 0.01–0.07 |
| Northern US [32] | 100–160 | 0.314 | N41-45˚ | 0.01–0.04 |
| Southern US [32] | 22–112 | 0.174 | N30-41˚ | 0.005–0.05 |
| *Europe* | | | | |
| Finland [34] | 52–93 | 0.462 | N60-70˚ | 0.004–0.015 |
| Denmark [30, 31] | 110 | 0.240 | N55-58˚ | 0.017–0.03 |
| British Isles [28] | 74–193 | 0.400 | N50-59˚ | 0.007–0.04 |
| France [27] | 32–65 | 0.111 | N44-50˚ | 0.01–0.04 |
| Sardinia [33] | 144–152 | 0.222 | N39-41˚ | 0.025–0.05 |
| Italy [33] | 38–90 | 0.145 | N38-46˚ | 0.02–0.05 |

* Per 100,000 population. The prevalence of MS for each region is taken from data provided in [42]. A range is given because, often, a range of estimates is available for a particular region.

† Estimates are presented as proband-wise concordance rates [26]. Sometimes concordance was reported as a pair-wise rate and, in these cases, the estimates have been converted into proband-wise rates assuming random sampling of twin-pairs [26]. Nevertheless, in at least some reports [e.g., 32], this assumption is almost certainly violated.

†† For the purposes of determining the probability of "genetic-susceptibility" {$P(G)$} in each region, we have taken: $P(MS) \approx 2^{*}(prevalence)$ – *see Text* – and we have adjusted the *MZ*-twin concordance rates using the Canadian data for differences between fraternal-twin and sibling concordance (*see Text*) $P(MS|IG_{MS}) = (2.9/5.4)^{*}P(MS|MZ_{MS})$. Finally, the range of values for $P(G)$ is taken both from the range of the prevalence data and also from the range provided by *Proposition #1* (*see Text*).

our estimate for $P(MS)$–*see #1 above*–it follows that:

$$0.022 = 0.003/0.134 \leq P(G) < 0.003/0.067 = 0.045$$

Consequently, by this analysis, only 4.5% or less of the general population (*Z*) could possibly be genetically susceptible to getting MS and the remainder of the population would have no possibility of getting this condition, regardless of their environmental experiences. Multiple reports from other MS-populations throughout Europe and North America yield very similar Upper Solution estimates for $P(G)$, which seems to be independent of latitude (*Table 4*).

Notably, we arrived at this estimate for {$P(MS|IG_{MS})$} by adjusting the observed value of {$P(MS|MZ_{MS})$} downward to account for the presumed impact of the shared *IU* and early post-natal environments of *MZ*-Twins (*see #2 above*). To do this, we estimated the magnitude of this impact from the increased recurrence risk in *DZ*-twins compared to that in non-twin siblings (*see Methods; see also #1, in S1 File*). Although, the Canadian data suggests a larger discrepancy between {$P(MS \mid DZ_{MS})$ and $P(MS \mid S_{MS})$} compared to other studies [27–34, 50], it is still possible that our adjustment is too small. Even so, there is a limit to how large any adjustment can be. Thus, from *Fig 3*, it must be the case that:

$$P(MS|IG_{MS}) > P(MS|S_{MS}) \approx 0.029$$

Otherwise, there would be no increased risk of MS in persons who have 100% of their genes in common and don't share their *IU* and post-natal environments compared to persons who have only 50% of their genes in common and also don't share their *IU* and post-natal

environments. Importantly, however, even in this case:

$$P(G) < 0.003/(0.029/2) = 0.21$$

Therefore, even using this extreme estimate, the large majority of the population (>79%) would have no chance of getting MS, regardless of their environmental exposures (*see Proposition #1*).

**4b. The Lower Solution.**   *Conclusions*: $\forall\{x<x'/2\}$: $0.025 \leq P(G) \leq 0.18$

$$\text{and}: \quad \forall\{x < x'/2\}: \quad 0.006 \leq P(G1) \leq 0.063$$

*Argument*: The considerations in *#4a* pertain only to an Upper Solution and the observations from Canada regarding recurrence risks for the gender partition in MS make it clear that the set {X} is, at least, bimodal (*see #5, below*). Moreover, given the magnitude of the gender imbalance in the (G) subset, it seems possible that the distribution of {X} might conform to a Lower Solution. Such a circumstance may increase the upper limit for genetic susceptibility to MS from the 4.5% estimated in *#4a* (*above*). Nevertheless, even in this case, there are constraints on possible solutions. For example, because we are assuming that sub-subsets (G1) and (G2) with significantly different expected penetrance values, considered separately, each conform to an Upper Solution (*see Methods*), the application of *Eq 1A* (*above*), together with the fact that $(x_1 > x'/2)$ –see Proposition #1, *above*–and with our observational estimates for P(MS) and (x') –see #1 & #2, *above*–indicates that:

$$P(MS) > P(G1)x_1 > P(G1)*(x'/2)$$

or, with substitution: $P(G1) < 0.003/0.067 = 0.045$

Consequently, using these estimates, no more than 4.5% of the population can possibly be in the (G1) subset. In addition, we undertook an analysis, which incorporated possible errors in these epidemiological observations. We then iteratively assigned, to each input parameter {g, p, x', r, s, & P(MS)}, values which spaned their entire plausible ranges, solved *Eqs 5a & 5b* (*see #3a, in S1 File*) for the Lower Solution using the different parameter combinations, and determined which combinations satisfied the constraints placed by the epidemiological observations (*see #3b, in S1 File*). From this analysis we conclude that:

$$\forall\{x < x'/2\}: \quad 0.025 \leq P(G) \leq 0.18$$

$$\text{and}: \quad \forall\{x < x'/2\}: \quad 0.006 \leq P(G1) \leq 0.063$$

Thus, although Lower Solutions exist for which, {P(G) = 1}, none of these solutions match both the constraints placed by the observed the values of {x', $x_1$', $x_2$', P(MS) & P(F|MS)} for the gender-partition and the requirement, when their expected penetrances are different, that (G1) and (G2) each conform to an Upper Solution (*see #3b, in S1 File; see also Table 3; Fig 3 & #5, below*). Indeed, this analysis demonstrated that:

$$\forall\{P(G) = 1\} : x_2' < 0.009$$

which is far removed from the actual observational data (*Table 3; Fig 3*). It seems, therefore, that the circumstance of {P(G) = 1} is excluded, even for Lower Solutions, in all but the most extreme distributional circumstances and, thus, for the majority of the population, developing MS is not possible. In earlier iterations of this analysis [3, 4, 51, 52], we defined the (G) subset differently–i.e., as $\forall G_i \in G: P(MS|G_i) \geq P(MS)$. Also, we note that, in the present analysis for Lower Solutions, our older definition effectively corresponds to defining only members of the (G1) subset as being genetically-susceptible to MS.

## 5. Genetic susceptibility in the gender partition–$P(F|G)$ & $P(M|G)$

*Conclusions*: 1. The set {X} has, at least, a bimodal distribution

 2. $0.145 \leq P(MS|F,G) \leq 0.187$

 3. $0.017 \leq P(MS|M,G) \leq 0.034$

 4. $0.18 \leq P(F|G) \leq 0.31$

 5. $4.3 \leq P(MS|F,G)/P(MS|M,G) \leq 8.7$

 6. $0.041 \leq P(G) < 0.073$

*Argument*: For ease of notation, the *Table 2* parameter abbreviations ($x, x', x_1, x_1', x_2,$ & $x_2'$) can be applied to the gender partition by defining both susceptible women and susceptible men such that: {(G1) = (F,G)} and: {(G2) = (M,G)}. The set {X} of penetrance values for members of the (G) subset is, at least, bimodal. Thus, from the data in *Fig 3*:

$$P(MS|F, MZ_{MS}) = 0.34 >> 0.067 = P(MS|M, MZ_{MS})$$

$$\chi^2 = 8.5; \ p = 0.0035$$

Because the sub-subsets (G1) and (G2) have significantly different expected pentrances, we assume that each, considered separately, conforms to the <u>Upper Solution</u> (*see Methods*). Therefore, from the estimated adjustments for the similar environment of *MZ*-twins for this partition (*see #1.1b, in S1 File*), together with the data in *Fig 3*, it follows that:

$$0.093 < x_1 = P(MS|F, G) \leq 0.187 \quad \text{(Eq 2A)}$$

$$\text{and}: \quad 0.017 < x_2 = P(MS|M, G) \leq 0.034 \quad \text{(Eq 2B)}$$

These possible ranges for men and women don't overlap. Therefore, for this partition, we have defined (*above*) the sub-subsets (G1) and (G2) correctly because: ($x_1 > x > x_2$) –*see Methods*. In this case: ($a > 1 > b$) and, as a consequence, $P(G1|G,MS)$ must be greater than $P(G1|G)$– *see #2a, in S1 File*. The proportion of MS patients who are women from *Table 3*; *Fig 3* is 66%. For the WTCCC data this number is 72%. From the study of Orton and colleagues [56] out of Canada, in the most recent epoch, the percentage of MS patients who are women is 76%. From a recent prevalence estimate for the United States [43], the percentage of women among MS patients is 74%. Using the data from *Table 3*; *Fig 3*, one possible upper limit for $P(F|G)$ is: $P(F|G) < P(F|G,MS) = 0.66$. Nevertheless, any such upper limit is too high. For example, using: $P(F) = P(M) = 0.5$, together with the other *Table 3*; *Fig 3* data and the above noted ranges for men and women, and from the definition of the (G) subset, we can estimate that:

$$P(MS, G|F) = P(MS|F) = \{P(F|MS) * P(MS)\}/P(F) = \{0.66 * 0.003\}/0.5 = 0.004$$

Because: $P(G|F) = P(MS,G|F)/P(MS|F,G)$

Therefore: $0.021 = 0.004/0.187 \leq P(G|F) < 0.004/0.093 = 0.043$

And similarly: $0.060 = 0.002/0.034 \leq P(G|M) < 0.002/0.017 = 0.118$

Therefore, the maximum proportion of women in the (G) subset (using the *Table 3*; *Fig 3* data) must be:

$$P(G|F)/P(G|M) = P(F|G)/P(M|G) < 0.043/0.060 = 0.717$$

$$\text{where}: \quad P(M|G) = 1/\{1 + P(F|G)/P(M|G)\} < 1/(1.717) = 0.582$$

$$\text{so that}: \quad P(F|G) < (0.717)*(0.582) = 0.42$$

In fact, the gender imbalance may be even greater than this (*see #4, in S1 File*). Thus, there are four serious concerns about undertaking any calculations that use the limits for ($x_1$ and: $x_2$) set forth by *Eqs 2A & #2B*, *above*. First, in making the above calculation, we are positing an extreme and tri-modal distribution for the set {$X$}–i.e., not the unimodal or bimodal distributions under primary consideration. Thus, this calculation, envisions a distribution, in which half of the women have a uniform penetrance of slightly greater than zero and the other half have a uniform penetrance of ($x_1'$) –i.e., women have the maximum variance possible–and, in which every man has exactly the same penetrance of ($x_2'$), which is intermediate between these two extreme penetrance groups of women–i.e., men have a zero variance.

Second, such an extreme distribution seems unlikely, especially for circumstances, in which partitioning the ($G$) subset by a different MS-associated characteristic–i.e., *HLA*-status (*see #6, below*)–doesn't even give a hint of the bimodal nature of {$X$}.

Third, it is not possible that the variance of penetrance values for the ($F,G$) subset to be at its maximum value. Thus, because, ($x_1' > x'$) –*see Table 3; Fig 3* –the maximum variance for the sub-subset ($F,G$) –($x_1'/2$)$^2$–exceeds the maximum total variance possible for the entire ($G$) sub-set– ($x'/2$)$^2$. Consequently, the lower limit for the value of ($x_1$) in *Eq 2A* –i.e., at its maximum possible variance–must be too low. And fourth, some of the maximum possible variance in the {$X$} set must be accounted for just by the separation of ($x_1$) from ($x_2$) –*see #4, in S1 File*.

Following the standard development of variance relationships [57], and taking each of these factors into account (*see #4, in S1 File*), including all solutions (either <u>Upper</u> or <u>Lower</u>), in which the penetrance values of ($G1$) and ($G2$) each follow an <u>Upper Solution</u>, leads to the conclusion that:

$$0.18 \leq P(F|G) \leq 0.31$$

$$4.3 \leq P(MS|F, G)/P(MS|M, G) \leq 8.7$$

$$\text{and that}: \quad 0.041 \leq P(G) < 0.073$$

Importantly, however, if the distribution of {$X$} follows an <u>Upper Solution</u>, those limits still apply (*see #4a*, *above*) although the somewhat different estimates for $P(G)$, in this circumstance, would need to be reconciled. Because the estimate derived from *Table 3* for the quantity {$P(MS|M,IG_{MS})$} is based on only two concordant twins, this seems likely to be the least reliable of any in the *Table*. Thus, if this estimated penetrance were doubled, there would still be an excess of men in the ($G$) subset such that:

$$0.31 \leq P(F|G) \leq 0.47$$

$$\text{but also}: \quad 0.026 \leq P(G) \leq 0.043$$

Consequently, an underestimate of {$P(MS|M,IG_{MS})$} would help with any such reconciliation. Similarly, considering only the possibility that the penetrance values of both ($G1$) and $G2$) are distributed in a unimodal manner would also help (*see #4, in S1 File*), as would an underestimate (from *Table 3; Fig 3*) for the proportion of women among MS patients (*see above, this section, see #8, below, & see #3b, in S1 File*).

Regardless, however, it seems clear not only that genetic susceptibility is rare in the population, even for <u>Lower Solutions</u>, but also that men are more likely than women to be genetically

susceptible to MS. At first pass, it might seem biologically improbable that men would be more likely than women to be in the genetically-susceptible subset ($G$). Thus, if membership in the ($G$) subset is envisioned as being due to an individual possessing a sufficient combination of some number of loci in a "susceptible state" [58], it is unclear how men could be more likely than women (or *vice versa*) to possess certain combinations and not others. This seems especially unlikely for circumstances, where one association study, specifically focused on the $X$-chromosome, failed to identify any susceptibility loci on this chromosome [7], where another large GWAS found that all but one of the 233 MS-associated loci were located on autosomal chromosomes [14], and where no major gender interaction term has been reported in the literature. Indeed, considering the different "risk" haplotypes in the *HLA* region identified in the WTCCC, men and women seem equally likely to be carriers [59]. Nevertheless, we can designate ($G_{ak}$) to represent each of the ($n$) autosomal genotypes ($k = 1,2,\ldots,n$) in the general population ($Z$) – i.e., omitting any specification of gender. In this circumstance, it is entirely possible that:

$$\forall G_{ak} \in Z : P(G_{ak}|M) = P(G_{ak}|F) = 0.5*P(G_{ak})$$

and, yet, for some specific autosomal genotypes to have the characteristic that:

$$P(G_{ak}, M) \in G \ \text{ and } : \ P(G_{ak}, F) \notin G$$

Indeed, such an explanation for the excess in susceptible men would fit well with the observation that the specific genetic combinations, which underlie susceptibility to MS, seem to be unique to each individual (*see #9, below; see also #7, in S1 File*). In addition, such a circumstance might also help to rationalize the finding that men seem to have a lower threshold of environmental exposure for developing MS compared to women (*see #7, below*).

## 6. Genetic susceptibility in the HLA partition–$P(G|H+)$ & $P(G|H-)$

***Conclusions*** :    $P(G|H+) \approx 3.35*P(G|H-)$

$$P(G|H+) \leq P(G)/P(H+) < 0.20$$

***Argument***: We will designate individuals who possess 1 or 2 copies of the Class II *HLA-DRB1\*15:01~HLA-DQB1\*06:02~a1* haplotype–i.e., the ($H+$) haplotype–as being members of the ($H+$) subset and those who possess 0 copies of this haplotype as being members of the ($H-$) subset. Some epidemiological studies only report *HLA-DRB1\*15* or *HLA~DRB1\*15:01* carrier status. Nevertheless, because, in the WTCCC, 93.4% of *HLA-DRB1\*15*-alleles are actually the *HLA-DRB1\*15:01* allele, and because 99% of *HLA-DRB1\*15:01* carriers also carry the full haplotype [60], each of these designations will be used interchangeably as ($H+$).

It is clear that ($H+$) status is considerably enriched in the MS population compared to controls. For example, in WTCCC controls {$P(H+) = 0.23$}, whereas in cases {$P(H+|MS) = 0.50$}. This enrichment of ($H+$) status in MS could occur in two ways (*see #5, in S1 File*). First, ($H+$) could make membership in the ($G$) subset more likely than it is for the ($H-$)-subset–i.e., it is due to an impact on the ratio of: $P(G|H+)/P(G|H-)$. Second, members of the ($G,H+$) subset may have a greater penetrance for MS than members of the ($G,H-$) subset–i.e., it is due to an impact on the ratio of: $P(MS|G,H+)/P(MS|G,H-)$. The available epidemiological data (*see #5, in S1 File*) suggests that the majority of enrichment is the due to the 1st of these two possible mechanisms and that:

$$P(G|H+) \approx 3.35*P(G|H-)$$

In addition, the observation (from the <u>Lower Solution</u>) that less than 7.3% of the population is genetically susceptible (*see #5; above*), together with the WTCCC observation that: $P(H+) = 0.23$, indicates that fewer than 32% (7.3/23) of ($H+$)-carriers are even genetically susceptible to MS. Indeed, taken together, the fact that only half of MS-patients are in the ($H+$) subset and the fact that this estimate for genetic susceptibility represents an upper bound for the <u>Lower Solution</u>, indicates that the actual percentage of ($H+$) carriers who are genetically susceptible must be far less than this 32% figure. Nevertheless, essentially all of the conserved extended haplotypes (*CEHs*) that carry ($H+$) – even those with a single representation in the WTCCC dataset – are associated with MS [60]. Therefore, it seems likely that all ($H+$)-carrying *CEHs* can contribute to genetic susceptibility. Despite this contribution, however, the majority of ($H+$) subset members have no chance whatsoever of developing MS. Therefore, at least with respect to the ($H+$)-carrying *CEHs*, genetic susceptibility to MS must result from the combined effect of ($H+$) together with the effects of certain other (as yet, unidentified) genetic factors (*see #7, in S1 File*). By itself, however, ($H+$) membership poses no MS-risk.

## 7. Environmental factors in MS

*Conclusions*: 1. Environmental factors are critical to MS pathogenesis

2. Susceptible women are more responsive than men to changes in the environmental conditions related to MS pathogenesis.

3. Compared to women, men have a lower threshold of environmental exposure at which they can develop MS

4. Currently, the environmental factors involved in MS pathogenesis are population-wide exposures.

5. Stochastic factors play an important role in MS pathogenesis

*Argument*: As noted in the *Methods*, we define ($E_T$) to be the prevailing environmental conditions (whatever these are) experienced by the population during some time-period ($T$). We also define ($E_i$) to be the specific environmental exposure, which is sufficient for MS to develop in the $i^{th}$ susceptible individual (however many events are involved, whenever these events need to act, and whatever these events might be)–i.e., both the events ($E_i$ and $G_i$) need to occur jointly in order for MS to develop in the ($i^{th}$) individual. Because genetic susceptibility is independent of the environmental conditions, the probability of a sufficient environmental exposure {$P(E)$} in the ($G$) subset at time-period ($T$) can be expressed as:

$$P(E) = P(E|G, E_T) = \sum_{i=1}^{m} P(E_i, G_i|G, E_T) = \sum_{i=1}^{m} P(G_i|G, E_T) * P(E_i|G_i, G, E_T)$$

$$\text{where}: \quad \forall G_i \in G : P(G_i|G, E_T) = P(G_i|G) = 1/m$$

When {$P(E) = 0$}, it is not possible for any susceptible person to experience an environment sufficient to cause MS. By contrast, when {$P(E) = 1$}, every susceptible person experiences an environment sufficient to cause MS. If there are some susceptible individuals, for whom <u>any</u> environmental experience is sufficient to cause MS (i.e., these individuals have "purely genetic" MS), then: $0 < P(E) \leq 1$ and thus, {$P(E) = 0$} cannot be observed. Importantly, those circumstances, in which {$P(E) = 0$}, only imply that, whatever environmental exposures take place (i.e., $E_T$), these are insufficient to cause MS in <u>anyone</u>. Regardless, considering the definitions

 

of both *P(E)* and the *(G)* subset (*see Methods*), it is clear that:

$$P(MS, G, E) = P(MS)$$

Notably, also, the above expression for *P(E)* explicitly incorporates the possibility that each genotype in *(G)* may require a unique set of environmental events in order for MS to develop in that individual. Nevertheless, despite this possibility, the existing epidemiological data suggests that many (or most) MS patients are responding to similar environmental events and, thus, any large variability in this regard is probably not a major factor in MS pathogenesis.

For example, despite the fact that every MS patient (except *MZ*-twins) has a unique combination of "states" at the (>*200*) susceptibility loci (*see #7, in S1 File*), the population-based data from Canada indicates that the change in general environmental conditions (whatever these are), which have taken place between the time periods of (1941–1945) and (1976–1980), have produced, at a minimum, a 32% increase in the prevalence of MS (*see #6d, in S1 File*). Moreover, because this increase has occurred world-wide and predominantly in women [3, 4, 51, 52, 56], the (*F:M*) sex ratio for MS in Canada has increased during every 5-year increment except one between these two time-periods [56]. Over the entire interval, the ratio has increased from 2.2 in (1941–1945) to 3.2 in (1976–1980). These changes are far too rapid to be genetically based.

It is conceivable that this observed sex-ratio change might be artifactual. For example, if women were more likely than men to have minimally symptomatic MS, then, with such patients now being diagnosed by our improved imaging and laboratory methods, women might represent a disproportionate number of these newly diagnosed cases. Alternatively, in earlier eras, vague symptoms of MS in women may have been written off as "non-organic" more often than they were in men. Nevertheless, four lines of evidence argue strongly against this change being an artifact. First, this increase in the sex ratio began before, and continued up to, the advent of modern imaging and laboratory methods [56]. Second, among asymptomatic individuals, incidentally, found to have MS by MRI, the (*F:M*) ratio is approximately the same as current estimates for symptomatic MS and 80% of the those with spinal cord lesions are women – i.e., those lesions having, by far, the greatest odds for progression to "clinical" MS [61]. Third, if (as seems likely), women have a higher threshold for developing MS than men, this would require the difference in exposure between the genders to be one of degree not one of kind (*see below, this section; see also #6e, in S1 File*). Finally, and most persuasively, the greater penetrance of MS in women is confirmed independently by the *MZ*-twin data (*see #5 above*). Consequently, the increase observed in the (*F:M*) sex ratio of Canada [56] almost certainly has an environmental basis.

In addition, a prior Epstein Barr viral (*EBV*) infection seems to be a prerequisite for most (or all) genotypes in *(G)* to develop MS [3, 4, 51, 52, 62–64]. Indeed, if (as suggested by these studies) a prior *EBV* infection occurs in 100% of MS cases, this would indicate that *EBV* exposure can be designated as a 'necessary factor' and, as such, must be part of the causal pathway leading to MS [51]. In addition, the likelihood that members of the *(G)* subset will develop MS seems to be influenced greatly by vitamin D deficiency, latitude, migration, and the *IU* environment [3, 4, 51, 52, 62–64]. Each of these additional observations also indicates that similar environmental changes can affect a large proportion of genetically susceptible individuals in a similar manner (i.e., contribute to MS pathogenesis).

Using the standard methods of survival analysis [65], we can define the cumulative survival $\{S(u)\}$ and failure $\{F(u)\}$ functions as well as the hazard-rate functions $\{h(u)\}$ and $\{g(u)\}$ for developing MS at different environmental exposures in "susceptible" men and women (respectively). These hazard-rate functions are assumed (initially) to be proportional. The

 

                                                

implications of non-proportionality are considered in the *in S1 File (#6e)* and in the legend of *Fig 4*. However, assuming proportionality, then:

$$g(u) = R*h(u) \ \text{ where}: \ \ u = P(E)$$

For men, we can transform exposure from $(u)$ units into $(a)$ units, first by defining $\{H(u)\}$ to be the definite integral of the hazard-function $\{h(u)\}$ from a $(u)$ level of exposure to a $(0)$ level of exposure and, second, by defining the $(a)$ units to be:

$$a = H(u) = \int_0^u h(u)du \ \text{ where}: \ \ da = h(u)du$$

Because these $(a)$ units are arbitrary, we can assign "1 unit" of environmental exposure in men to be the difference in exposure level between any two time points (e.g., $a_1$ and $a_2$) such that:

$$a_2 - a_1 = 1$$

For women, we can similarly transform exposure into a different scale of so-called "apparent" exposure units $(a^{app})$ such that:

$$a^{app} = R*a$$

and where we now define "1 unit" of environmental exposure (on this scale) as:

$$a_2{}^{app} - a_1{}^{app} = 1$$

The choice of which gender (*men* or *women*) to assign to which scale is completely arbitrary.

A standard derivation from survival analysis methods [65], demonstrates that the survival curves are exponential with respect to their hazard functions.

Thus, for men: $\ln[S(u)] = -\int_0^u h(u)du \ \text{ or } \ : \ln[S(a)] = -\int_0^a da = -a$

and, for women:

$\ln[S(u)] = -\int_0^u R*h(u)du \ \text{ or}: \ \ln[S(a)] = -\int_0^a R*da = -Ra = -a^{app} = \ln[S(a^{app})]$

So that, for men: $S(a) = e^{-a}$ and : $F(a) = 1 - e^{-a}$

and, for women: $S(a^{app}) = e^{-a^{app}}$ and : $F(a^{app}) = 1 - e^{-a^{app}}$

In considering the probability of failure (i.e., of developing MS), we will use subscripts (*1*) and (*2*) to denote the failure probabilities and the values of other parameters at the 1st and 2nd time-periods respectively. Importantly, unlike true survival (where everyone fails given a sufficient amount of time), the probability of developing MS may not become 100% as the probability of a sufficient environmental exposure increases to $\{P(E) = 1\}$. Moreover, the limiting value for the cumulative probability of developing MS in men (*c*) need not be the same as that in women (*d*). However, because the new definition of the subset (*G*) differs from earlier iterations of our analysis [3, 4, 51, 52], the environmental exposure at which the development of MS becomes possible (i.e., the threshold) must occur at $\{P(E) = 0\}$ for, at least, one of these two sub-subsets–provided that this exposure level is possible for either one or both of these 2 gender subgroups (*see Fig 4, and above*).

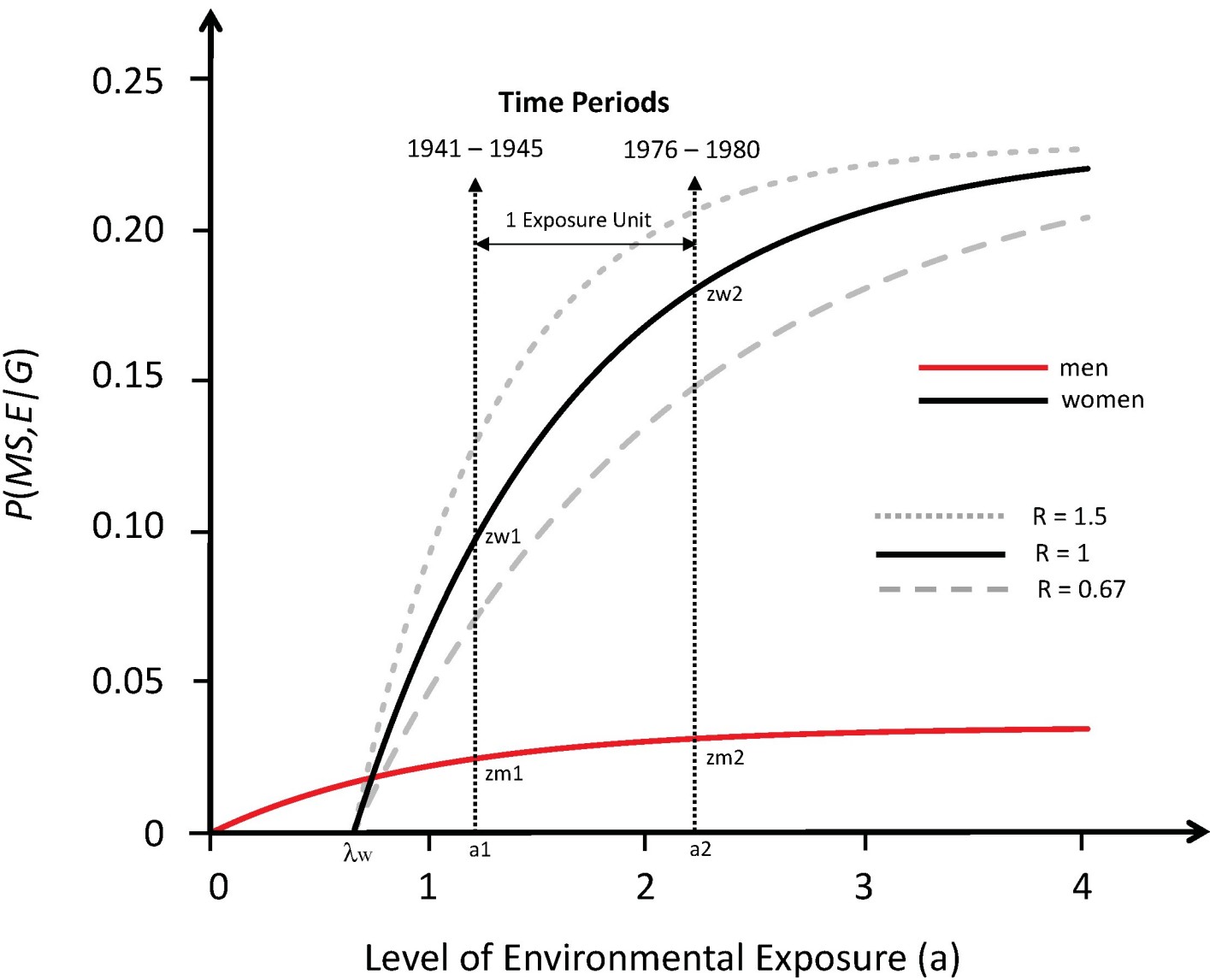

**Fig 4. Response curves for the likelihood of developing MS in genetically susceptible men and women with an increasing probability of a sufficient environmental exposure {$P(E)$}, assuming proportional hazards (*see #7*).** Response curves are derived from the change in the (*F:M*) sex-ratio over time in Canada [56] and using the estimates for $P(MS)_2$ & $P(G)$ estimated in the *Text*. The probability of getting MS in a genetically-susceptible individual–i.e., $P(MS,E|G)$ –is shown on the *y-axis*. The exposure level {$P(E)$} for the population is shown on the *x-axis* using transformed "exposure units" (*a*)–*see #7*, *Text*. Labels for points $Zw = P(MS,E|G,F)$ and $Zm = P(MS,E|G, M)$ are provided at time-points (*1*) and (*2*). One "exposure unit" is defined arbitrarily as: ($a_2$–$a_1$) for men and ($a_2^{app}$–$a_1^{app}$) for women (*see #7*). Solid-line plots have been constructed using the known values of ($Zw_2$) and ($Zm_2$), estimating that: $P(F|MS)_2 = 0.66$ (*Table 3*; *Fig 3*), together with the estimates of: {$C = 0.6$}, {$R = 1$}, {$P(G) = 0.044$} & {$P(F|G) = 0.25$}– the latter two parameters being valued in the middle of their predicted ranges (*see*, *Text*). The limiting values for ($Zm$) and ($Zw$) are: ($c = 0.035$) and ($d = 0.228$), respectively. Increasing the estimate of $P(F|G)$ will reduce the separation of the response curves by lowering the plateau for women and raising it for men; increasing the estimate of $P(F|MS)_2$ will increase the separation of between the curves in men from women for the opposite reason; increasing the estimate of $P(G)$ will reduce the plateaus of both response curves. Response curves for women under conditions ($R = 0.67$) and: ($R = 1.5$) are also depicted and are shown in grey lines (dashed and dotted, respectively). Changes to the value of (*C*) will slightly alter the units of the *y-axis*. As seen in the *Figure*, men have a lower threshold for developing MS compared to women (*see #7*, *Text*), and changes to the value of (*R*) alter how quickly the curves reach their plateau (limit). If the hazard is not proportional, for women, each of the points ($Zw_1$, $Zw_2$, $Zm_1$, and $Zm_2$) would be the same as depicted for ($R = 1$), although the scale of the *x-axis* for the two exponential curves would be transformed non-linearly and, thus, the response-curve in men could not be plotted on the same graph as women. Moreover, the *x-intercept* for the curve in women would be at ($a^{app} = \lambda w = 0$). Nevertheless, the limiting values (*c*) and (*d*) would be unchanged and, under any circumstances, women (relative to men) would still be seen to exhibit a greater responsiveness to those changes in environmental exposure, which have taken place between the two time-periods. If some cases of MS were "purely genetic" (i.e., $P(E|M) = 0$, or $P(E|F) = 0$ or both were not possible), this could elevate the zero-point on *y-axis* for "environmental" MS to the intersection of the curves for men and women (*see Text*) and this would make the threshold difference then disappear (i.e., $\lambda = 0$) –*see #7*, *Text*.

From these definitions, the failure probability for susceptible women ($Zw$) and men ($Zm$) at the 1$^{st}$ time period is:

$$F(a^{app})_1 = Zw_1 = P(MS, E|G, F)_1 = d*\{1 - e^{-a1^{app}}\} \qquad (women)$$

and :
$$F(a)_1 = Zm_1 = P(MS, E|G, M)_1 = c*\{1 - e^{-a1}\} \qquad (men)$$

By our definitions of "1 exposure unit", these equations, at the 2$^{nd}$ time point, become:

$$F(a^{app})_2 = Zw_2 = P(MS, E|G, F)_2 = d*\{1 - e^{-a1^{app}+1}\} \qquad (women)$$

and :
$$F(a)_2 = Zm_2 = P(MS, E|G, M)_2 = c*\{1 - e^{-a1+1}\} \qquad (men)$$

Because the observations at time-periods (1) and (2) represent two points on the exponential response curves for both women and men, and because any two points on an exponential curve defines the curve (both uniquely and completely), we can use the observations regarding the ($F$:$M$) sex-ratio change over time in Canada [56], to derive and construct these two response curves.

Thus, from the definition of {$P(E)$} and using: {$P(G,F) = P(F|G)^**P(G) = 0.25*0.044 = 0.011$ & $P(F|MS)_2 = 0.66$ – i.e., $P(F|G)$ and $P(G)$ are in the middle of their estimated ranges (*see #5 above; see also #4, in* S1 File) and $P(F|MS)_2$ is taken from Table 3 – we can estimate the values of ($Zw_2$) and ($Zm_2$) as:

$$Zw_2 = P(MS, E|G, F)_2 = P(MS|G, F)_2 = P(F|MS)_2*P(MS)_2/P(G, F) = 0.180$$

and:

$$Zm_2 = P(MS, E|G, M)_2 = P(MS|G, M)_2 = P(M|MS)_2*P(MS)_2/P(G, M) = 0.031$$

Moreover, as demonstrated in the *in* S1 File *(#6a & #6d)*, we define the term ($C$) such that:

$$C = P(MS)_1/P(MS)_2$$

and, thereby, re-express ($Zw_1$) and ($Zm_1$) in terms of ($Zw_2$) and ($Zm_2$) such that:

$$Zw_1 = \{P(F|MS)_1/P(F|MS)_2\}*C*Zw_2$$

and :
$$Zm_1 = \{P(M|MS)_1/P(M|MS)_2\}*C*Zm_2$$

where :
$$C < P(M|MS)_2/P(M|MS)_1 = 0.238/0.313 = 0.76$$

And, thus :
$$P(MS)_2 = (1/C)*P(MS)_1 = 1.32*P(MS)_1$$

Consequently, based on the population data from Canada, the prevalence of MS must have increased by more than 32% between these two time periods.

Finally (*see #6b, in* S1 File), we can estimate the value of both (*c*) and (*d*) as:

$$c = (Zm_2)*\{1 - [P(M|MS)_1/P(M|MS)_2]*C*e^{-1}\}/(1 - e^{-1})$$

and :
$$d = (Zw_2)*\{1 - [P(F|MS)_1/P(F|MS)_2]*C*e^{-1}\}/(1 - e^{-1})$$

Thus, using the observed change in the (*F*:*M*) sex-ratio over time in Canada, together with our estimates for $P(G)$ and $P(F|G)$, we have all the data needed to construct the complete response curves for the probability of developing MS with a changing environmental exposure

in genetically susceptible women and men (*Fig 4*). What these curves make clear is that both $P(E)$ and $P(MS)$ are changing over time, which indicates that specific environmental conditions, in addition to specific susceptible genetic combinations, are necessary for MS to develop. Thus, MS develops when the right genetic constitution is exposed to the right environmental conditions (i.e., it is fundamentally due to a gene-environment interaction).

Because, the scales for the response-curves for women and men are initially assumed to be proportional they can be plotted on the graph (*see Fig 4; see also #6e, in S1 File*) and, when this is done, the threshold (*x-intercept*) occurs at $\{(a,Zm) = (\lambda m,0)\}$ for men and $\{(a,Zw) = (\lambda w,0)\}$ for women. By the definitions of $(E)$ and $(a)$, one of these thresholds must occur at $\{(a,Z) = (0,0)\}$ –provided this exposure level is possible (*see Fig 4 & above, same section*). However, because these thresholds need not be the same, we define the difference in threshold between women and men as: $(\lambda = \lambda w - \lambda m)$ such that, if women have a higher threshold than men: $(\lambda > 0)$. However, as noted (*above*), the $(a)$ scale (for men) may be different than the $(a^{app})$ scale for women so that in order to plot them on the same graph requires the conversion of $(a^{app})$ units into $(a)$ units–*see #6c, in S1 File*.

Three final points are also worth making. First, because, as demonstrated in the (*#6c*) *in S1 File*, $(\lambda w)$ is independent of $R$, we can use the condition of $(R = 1)$ to evaluate $(\lambda w)$. In this circumstance, these exponential equations can be re-arranged (*#6c, in S1 File*) to yield:

$$\lambda = \ln\{[1 - Zw_2/d]/[1 - Zm2/c]\}$$

Consequently, basic epidemiologic data can be used to determine the difference in threshold $(\lambda)$ that exists between women and men. As demonstrated in the *in S1 File (#6c)*, this leads to the two conclusions that:

$$\forall C > 0.50: \quad 0.37 < \lambda < 4.67; \text{ and}: \quad \forall C > 0: \lambda > 0$$

Moreover, the value of $(\lambda)$ depends only upon the value of $(C)$ and the sex-ratio change over time so that, if the hazards are proportional, men must have a lower threshold for developing MS compared to women (*Fig 4; see also #6c, in S1 File*). A lower threshold in men is also suggested by a report from Europe and the United States [66], which found that prior to 1922 men accounted for 58% of the MS cases (*Table 5*). By our definition of $P(E)$, these thresholds indicate the exposure, at which MS becomes possible. If women required a fundamentally different kind of exposure than men, it would be very hard to rationalize a difference in threshold because, in such a circumstance, in some environments, women would be more likely and, in other environments, less likely than men to receive the correct exposure. Rather, a difference in threshold implies that men and women are responding to similar events but that men require a less extreme degree of exposure in order to develop MS. For example, perhaps, susceptible men develop MS with a lesser degree of vitamin D deficiency or with EBV infection occurring over a broader age-range compared to susceptible women. {*NB: even if there were no threshold difference, proportionality, by itself, would suggest that difference in exposure was one of degree but not kind.*}

Alternatively, there may an environment-gender interaction such that susceptible men, in any given environment (i.e., $E_T$), are more likely to experience a sufficient exposure than susceptible women. For example, perhaps men are more likely to engage in "risky" behaviors compared to women, or that they are more likely to be "sun-averse" than women. Having said this, however, it is not clear how (or whether) "individual" differences in behavior (even if they are biologically driven) could lead to a "population-level" difference in threshold (*Fig 4*). More likely, any such interactions would have to be related to physiological differences between the genders.

**Table 5. Sex distribution of multiple sclerosis cases reported prior to 1922[*].**

|  | Europe (Historical) | United States (Historical) | United States (Wechsler Series) | Total |
|---|---|---|---|---|
| **Men** | 658 (58%) | 99 (60%) | 117 (59%) | 874 (58%) |
| **Women** | 484 (42%) | 67 (40%) | 80 (41%) | 631 (42%) |
| **Total** | 1,142 (100%) | 166 (100%) | 197 (100%) | 1,505 (100%) |

[*] Data from: Wechsler IS [66]. Historical cases were reported in the medical literature, for the most part, prior to 1903, whereas Wechsler's series was drawn from the Mount Sinai Hospital, the Montefiore Hospital, and the Vanderbilt Clinic (1912–1921).

Another possibility is that a small percentage of MS patients (in men or women or both), have "purely genetic "MS, whereby <u>any</u> environment is sufficient to cause MS, given their genotypes. Such a circumstance renders the points ($\lambda w = 0$), or ($\lambda m = 0$) or both unobservable, as drawn in *Fig 4* (*see above; same section*). For example, in *Fig 4*, if ~1.8% of both susceptible men and women had "purely genetic" MS, this would raise the zero point of the *y-axis* for "environmental" MS such that this threshold difference would disappear (i.e., $\lambda = 0$) and both men and women would begin their "environmental" response at the new (0, 0) point in *Fig 4* –i.e., at the point of intersection of the two curves. The same would be true if only men had this percentage of "purely genetic" MS except that, in this case, men would begin their "environmental" response at the point of intersection–i.e., at ($Zm = 0.018$), whereas women would begin at ($Zw = 0$), which would define the the new onset point (0,0). If both had more than this percentage (or other combinations), the exact relationship between the curves at the start would change but, depending upon the exact situation, there still could be no difference in the threshold for "environmental" MS (*Fig 4*). Clearly, this example only applies to the specific conditions of *Fig 4*. Nevertheless, because ($\lambda > 0$) and because, at every exposure at or below the exposure at the point of intersection: ($0 < Zm < Zm_1$), in this circumstance, only a small amount of "purely genetic" MS would be necessary to eliminate the threshold difference for every condition (*see also #8 below*).

Second, we note that: {$P(MS|G,E,M) = c$}, so that ($Zm2$) can be re-expressed as:

$$Zm_2 = P(MS, E|G, M)_2 = P(MS|E, G, M)_2 * P(E|G, M)_2 = c * P(E|G, M)_2$$

This equation can be rearranged to yield: $P(E|G,M)_2 = Zm_2/c$
From *above*: $Zm_2/c = (1 - e^{-1})/\{1 - [P(M|MS)_1/P(M|MS)_2]^* C^* e^{-1}\}$
Therefore, in men $\forall C > 0.5$: $P(E|G,M)_2 > 0.83$
And, similarly, in women: $\forall C > 0.5$: $P(E|G,F)_2 > 0.76$

These results strongly suggest that the relevant environmental exposures (especially when these are multiple) are currently occurring at population-wide levels. For example, if three, equally likely and independent, environmental events ($EE_1$, $EE_2$, and $EE_3$)–possibly sequential [51, 52]–were sufficient to produce MS in a susceptible individual, then:

$$P(E) = P(EE_1) * P(EE_2) * P(EE_3) = P(EE_1)^3 = P(EE_2)^3 = P(EE_3)^3 = 0.83$$

$$\text{or}: \quad P(EE_1) = P(EE_2) = P(EE_3) = (0.83)^{1/3} = 0.94$$

so that, under the stated circumstances, more than 94% of the population would experience each environmental event. Such a conclusion is fully consistent with the same conclusion reached from studies in adopted individuals, in siblings and half-siblings raised together or apart, in conjugal couples, and in brothers and sisters of different birth order, which have generally indicated that MS-risk is unaffected by the micro-environments of families but, rather, results from population-wide exposures [67–73].

And third, it is clear that both of these response curves plateau well below 100% failure, especially in men (*Fig 4*). Therefore, there must be stochastic processes that partially determine whether a susceptible individual with a sufficient environmental exposure will actually develop disease (*see #9, below*).

## 8. The future fate of $P(MS \mid IG_{MS})$

***Conclusions:*** 1. $\lim_{P(E) \to 1} P(MS, E|F, IG_{MS}) \geq P(MS|F, MZ_{MS})$

 2. $\lim_{P(E) \to 1} P(MS, E|M, IG_{MS}) \geq P(MS|M, MZ_{MS})$

***Argument:*** As a sufficient environmental exposure $\{P(E)\}$ becomes more likely, the quantity $\{P(MS \mid IGMS)\}$ will, of necessity, change. Earlier, we described this term as having removed the impact of the shared *IU* and certain (especially early) post-natal environments of *MZ*-twins. This description, however, is not quite accurate. For example, we can break down a "sufficient" environmental exposure (*see #1, in S1 File*) into those factors that are shared exclusively by *MZ*-twins ($E1$), those factors that are shared by the population generally ($E2$), and those factors that shared exclusively within the family micro-environment ($E3$). As noted above, however, the family micro-environment seems not to have any impact on the likelihood of MS [67–73]. In this circumstance, assuming only factors ($E1$ and $E2$) are necessary for a sufficient exposure, then:

$$P(MS, E) = P(MS, E_1, E_2, E_3) = P(MS, E_1, E_2) = P(E_1) * P(MS, E_2|E_1)$$

and : $\qquad P(MS, E|MZ_{MS}) = P(MS, E_1, E_2|MZ_{MS}) = P(E_1|MZ_{MS}) * P(MS, E_2|E_1, MZ_{MS})$

If an individual's identical twin is known to have MS, it is likely that this individual, also, has experienced a "sufficient" ($E1$) exposure.

Conceived of in this way, the term $\{P(MS \mid IGMS)\}$ can be rewritten as:

$$P(MS|IG_{MS}) = P(E_1) * P(MS, E_2|E_1, MZ_{MS})$$

and the adjusted penetrance $\{P(MS|IGMS)\}$ hasn't really "removed" the impact of these environmental similarities. Rather, $\{P(E_1|MZ_{MS})\}$ has simply been reset to its population level $\{P(E_1)\}$. Because *MZ*-twins share both identical genotypes and the same *IU* and certain post-natal environments, we expect that: $P(E_1|MZ_{MS}) = 1$. Consequently, as $\{P(E_1)\}$ increases in the population to:

$$P(E_1) = P(E_1|MZ_{MS}) = 1$$

the term $\{P(MS \mid IGMS)\}$ will approach, and ultimately reach:

$$P(MS|IG_{MS}) = P(MS|MZ_{MS})$$

In this case, therefore, the limiting value for $\{P(MS,E|G)\}$ in men (***c***) and women (***d***)–*see #7 above; see also Fig 4* –must conform to the constraints of:

$$c = P(MS|E, M, G) \geq P(MS|M, MZ_{MS})$$
$$d = P(MS|E, F, G) \geq P(MS|F, MZ_{MS})$$

The reason for the inequality is that, in those circumstance where:

$$P(E) = P(E_1, E_2) = P(E_1) * P(E_2|E_1) = 1$$

it must be that both: $P(E_1) = 1$ and: $P(E_2|E_1) = 1$. Naturally, the fact that $\{P(E_1)\}$ has increased to unity does not guarantee that $\{P(E_2|E_1)\}$ has done the same, so that the limiting value for $P(MS,E|G)$ may be greater than $P(MS|MZ_{MS})$.

Nevertheless, if it is currently true (*see #7 above*), that:

$$P(E) > 0.76 \quad \text{and, thus}: \quad P(E_2 \mid E_1) > 0.76;$$

then it must also true that: $c \approx P(MS|M, MZ_{MS})$ and: $d \approx P(MS|F, MZ_{MS})$}

Regardless, however, the depicted curves (*Fig 4*) must be inaccurate because, in the Figure:

$$c = 0.035 < 0.067 = P(MS|M, MZ_{MS})$$

$$\text{and}: \qquad d = 0.228 < 0.34 = P(MS|F, MZ_{MS})$$

Clearly there are several variables that can be adjusted {$C$, $R$, $P(G)$, $P(F|G)$ and $P(MS)$} to match the values for both (***c***) and (***d***) with these observed *MZ*-twin concordance rates. Therefore, we iteratively considered various combinations of these variables and determined which of those combinations matched these constraints. Specifically, we considered the plausible variables ranges of: ($0.25 \leq C \leq 0.75$), ($0.20 \leq R \leq 5.0$), ($0.001 \leq P(G) \leq 1.0$), ($0.18 \leq P(F|G) \leq 0.70$), and ($0.002 \leq P(MS) \leq 0.006$), and further required that the estimates for (***c***) and (***d***) be within (± 15%) of the observed values for their proband-wise *MZ*-twin concordance rates (*Table 3*; *Fig 3*). In this analysis, we found numerous combinations, which matched these constraints. The solution space covered by these matching combinations included the full range of possibilities for the parameters of ($C$) and ($R$). By contrast, the ranges for both $P(F|G)$ and $P(G)$ were restricted: {$0.33 \leq P(F|G) \leq 0.5$} and {$0.02 \leq P(G) \leq 0.055$}. This restricted range for $P(G)$ fits, generally, within the framework developed previously and confirms the conclusion that developing MS is not a possibility for a large majority of the population (*see #4a & #4b above*). Similarly, this analysis confirms that women are less likely than men to be in the ($G$) subset, although the estimated range for $P(F|G)$ is somewhat higher than the ranges developed previously (*see #5 above; see also #4 in S1 File*). As discussed in the *in S1 File* (#4), however, this could relate to an underestimate for the parameter {$P(MS|M, MZ_{MS})$}, which is based upon only 2 observations of concordant male *MZ*-twins (*Table 3*).

Also, the 5 potential solutions for which: {$P(MS) \leq 0.003$} accounted for 11% of the total matching combinations. By contrast, the 5 potential solutions for which: {$P(MS) \geq 0.004$} accounted for 79% of the total. This circumstance suggests that we are under-estimating $P(MS)$ when using the observed disease prevalence in the general population ($Z$). Indeed, several autopsy studies have indicated that the prevalence of undiagnosed (pathological) MS is ~0.1% [46–51]. Thus, with minimally symptomatic (or asymptomatic) MS occurring in as many 0.1% of the population, this could potentially increase the estimated $P(MS)$ by as much as 50–100%. Although, such diagnostic errors are probably less common in the modern era, many minimally symptomatic (or asymptomatic) patients are still being undiagnosed during life [59]. Moreover, any such under-ascertainment is likely to be less for *MZ*-twins, *DZ*-twins, and siblings than in the general population. For example, an initially unaffected twin or non-twin sibling of a patient with MS will, almost certainly, be more carefully monitored for possible MS symptoms (i.e., for minimally symptomatic presentations) than will an individual in the general population. In such a circumstance, these diagnostic failures will be fewer in the ($MZ_{MS}$), ($DZ_{MS}$), and ($S_{MS}$) populations than in the general population and the *MZ*-twin concordance rates will, thus, provide a more accurate reflection of the maximum likelihood of getting MS {i.e., $P(MS|G, E_1, E_2)$} than will those estimates of $P(MS)$ derived from the *MS*-prevalence in the general population. Such a circumstance would help to account for this apparent discrepancy.

## 9. Missing heritability?

*Conclusions*: 1. Both "genetic" and "environmental" factors are necessary for MS

expression; Neither alone are sufficient.

2. A large portion of the "causal pathway" to MS is stochastic

3. There is no need to invoke any "missing heritability" in MS

***Argument***: Only a small proportion of the population seems to be genetically susceptible to developing MS, which implies that MS is a genetic disorder. In addition, a suitable environmental exposure, like a suitable genetic constitution, is also a necessary part of MS pathogenesis. Despite this, however, the combination of a susceptible genotype together with a sufficient environmental exposure, does not invariably lead to the disease of MS and, in fact, the response curves in both women and especially men plateau well below 100% (*Fig 4*), even when everyone receives an environmental exposure suitable for their particular genotype–i.e., when {$P(E)$ =1}. This variance in the likelihood of getting MS for certain susceptible genotypes cannot be attributed to unidentified environmental conditions because the definition of the term {$P(E)$} – *see #7 above* – explicitly includes all such factors, both if they are known (or suspected) and also if they are completely unknown. Therefore, a large portion of the overall variance in MS disease-expression must be due to stochastic processes.

In this context, dividing the total variance in disease expression into genetic and environmental components, at least for MS, mischaracterizes the situation. This has important implications regarding current estimates for the "missing hereditability" in MS [74–76]. First, as noted above, a large portion of the variability in MS expression must be due to stochastic processes that are neither environmental nor genetic. And second, specific gene-gene combinations (likely unique to individuals or very small groups of individuals) must underlie genetic susceptibility to MS (*see #6 above; see also #7, in S1 File*). Thus, with over 200 MS-associated loci [14], each (potentially) having more than one "susceptible state" (e.g., the *MHC*), the number of possible combinations of states at these loci is so huge that, almost certainly, everyone (except *MZ*-twins) possesses a unique combination of these "susceptible states" (*see #7, in S1 File*). Indeed, considering (*H+*)-status together with only the first 102 of these MS-associated *SNPs* [13], everyone (including both cases and controls) in the WTCCC population does, in fact, possess a unique combination (*#7, in S1 File*). Consequently, if only a few such combinations are members of the (*G*) subset, even among those combinations that are quite similar to each other (*see #6 above; see also #7, in S1 File*), then there are more than enough genetic associations already identified to account fully for (*G*) subset membership. Naturally, many more loci may yet be identified, although positing their existence is unnecessary.

Alternatively, if "missing heritability" is only meant to imply that our genetic model cannot predict accurately the occurrence of MS, then, indeed, almost all of the heritability of MS remains unexplained. Thus, the environmental factors, the actual (as opposed to associated) genetic factors involved in causing disease, the necessary gene-gene combinations, the various gene-environment interactions, and the stochastic factors–all of which contribute importantly to whether MS can, or will, develop in a specific individual–are poorly understood, thereby making any accurate prediction of MS occurrence impossible at present.

## Discussion

The present analysis provides considerable insight to the nature and basis of susceptibility to MS and to the role of genetic determinants in polygenic diseases. Firstly, we establish that, fundamentally, MS pathogenesis requires both a genetic predisposition and a sufficient environmental exposure. Moreover, only a fraction of the population (less than 7.3%) is genetically-susceptible. Thus, more than 92.7% of the population has no chance of developing MS, regardless of the environmental conditions that these individuals experience. Thus, the correct genetic make-up is essential for disease pathogenesis. The basis of this genetic susceptibility,

however, is complex. Single genes or single haplotypes do not contribute much. For example, in MS, the Class II *HLA-DRB1\*15:01~HLA-DQB1\*06:02~a1*, or (*H+*), haplotype is the genetic trait with the largest (by far) disease-association of any in the genome (for the WTCCC: *OR* = 3.28; $p \ll 10^{-300}$). Nevertheless, despite this strong association more (and, likely, far more) than 68% of individuals who carry this haplotype have no MS-risk whatsoever. In this circumstance, it must be that genetic susceptibility depends upon the possession of this haplotype in combination with other genetic traits. Notably, this haplotype is only a part of much larger *CEHs*, which span the entire *MHC* region [23, 24]. Even considering the large number and variety of these highly selected *CEHs*, however, genetic susceptibility cannot be explained on the basis of the state of the *MHC*. Despite a significant variability in the observed disease-association among the different (*H+*)-carrying *CEHs*, every such *CEH* (regardless of its rarity) seems to be strongly MS-associated [23, 24].

In addition, it seems clear that, although certain genetic combinations increase the likelihood of (*G*) subset membership, the actual combinations that do this are quite heterogeneous, and only a small proportion of genetically susceptible individuals (who actually develop MS) share even the same 4-locus genetic combination (*see #7, in S1 File*). These observations also suggest that susceptibility to MS, although genetically based, is idiosyncratic.

Despite the conclusion that MS is genetic, however, MS is equally an environmental disease. Specific environmental exposures are also necessary for disease-pathogenesis. Indeed, the fact that there has been a marked recent increase in both MS-prevalence and the (*F:M*) sex-ratio, indicates that a sufficient environmental exposure is required for MS to develop (*Fig 4*). If a person is not exposed to a sufficient environment, they cannot get MS, regardless of their genetic make-up. However, neither environment nor genetics alone is sufficient. Rather, MS is due to an interaction between the two.

Several environmental events, probably sequential, seem necessary for MS to develop in a genetically susceptible individual [3, 4, 51, 52, 62–64]. The first environmental event, as discussed previously [51], is one that occurs during *IU* or early post-natal period. Support for such a factor comes from the discrepancy in recurrence-rates between twin and non-twin siblings, from the fact that concordant half-twins are twice as likely to share the mother than the father, and from the periodic, circa-annum, effect that month-of-birth has on the subsequent likelihood of developing MS [51]. In the northern hemisphere, this periodicity to MS-susceptibility peaks just before the summer months and dips to its nadir just before winter and this pattern is inverted southern hemisphere [51]. Each of these three observations implicates an environmental event, involved in MS pathogenesis, that is occurring near birth [51]. The evidence for a circa-annum periodicity to susceptibility suggests that this event is coupled to the solar cycle [51].

A second environmental event is implied by the published migration data [51]. Thus, when an individual relocates (prior to ~15 years of age) from an area of high-prevalence to an area of low-prevalence (or *vice versa*), their MS risk is similar to that of the area to which they moved. By contrast, when they make the same relocation after this time, their MS risk seems to remain that of the area from which they moved. These observations implicate an environmental event, involved in MS-pathogenesis, which occurs at or around puberty [51]. And third, the clinical onset of MS generally occurs long after the first and second events have already taken place (*Fig 1*), suggesting that one or more additional environmental events are also required for clinical MS to develop.

Naturally, there is no guarantee that the environmental events, which are sufficient to cause MS in one person, are the same as those that are sufficient in another. Nevertheless, those factors or events, which have been implicated in MS-pathogenesis so far, appear to affect a large proportion of susceptible individuals in a similar manner. Thus, the fact that we even have

evidence for the first two factors (*as described above*) suggests this. In addition, a prior Epstein Barr viral (*EBV*) infection has been strongly linked to MS, especially when this infection results in symptomatic mononucleosis. Indeed, such an infection prior to clinical onset occurs in ~100% of MS cases [3, 4, 51, 52, 62–64] and, if this is the case, this would indicate that *EBV* exposure is a 'necessary factor' in the causal pathway leading to MS [51]. Finally, there is a considerable amount of circumstantial evidence, which suggests a role for vitamin D deficiency in this causal pathway [51].

However, even when the correct genetic background occurs together with an environmental exposure sufficient to cause MS in someone of that background, more than 50% of such individuals will still not develop clinical disease. Some of these individuals, no doubt, will have subclinical disease [46–49, 61]. However, although such a circumstance will increase our estimate of $\{P(MS)\}$ by as much as 50–100%, this is still insufficient to get the plateaus of the response curves (*Fig 4*) to exceed the 50% mark. In men (who have a plateau significantly lower than that of women), this conclusion is even more evident (*Fig 4*). Consequently, because a sufficient environmental exposure has been defined broadly (to include both factors that are known or suspected as well as factors that are completely unknown), the fact that some individuals with the proper combination of genes and environment still fail to develop disease, indicates that stochastic processes are also involved in disease-pathogenesis.

And finally, it is worth noting that the nature of genetic susceptibility developed in this manuscript is applicable to a wide range of other complex polygenetic disorders such as type-1 diabetes mellitus, celiac disease, and rheumatoid arthritis. Indeed, based solely upon *Proposition #1*, if the proband-wise *MZ*-twin concordance rate, for any disease, greatly exceeds the prevalence of disease in the general population, then only a tiny fraction of the population has any possibility of getting the illness. Moreover, any disease for which the proband-wise *MZ*-twin concordance rate is substantially less than 100% must, in addition to genetic susceptibility, include environmental factors, stochastic factors, or both in the causal pathway leading to the disease.

## Supporting information

**S1 File.**
(PDF)

## Author Contributions

**Conceptualization:** Douglas S. Goodin.

**Formal analysis:** Douglas S. Goodin.

**Methodology:** Douglas S. Goodin, Pouya Khankhanian, Pierre-Antoine Gourraud, Nicolas Vince.

**Visualization:** Douglas S. Goodin.

**Writing – original draft:** Douglas S. Goodin.

**Writing – review & editing:** Pouya Khankhanian, Pierre-Antoine Gourraud, Nicolas Vince.

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
