## [Decision Letter · Decision Letter 0]

30 Dec 2020

PONE-D-20-25332

The Nature of Genetic Susceptibility to Multiple Sclerosis

PLOS ONE

Dear Dr. Goodin,

Thank you for submitting your manuscript to PLOS ONE. After careful consideration, we feel that it has merit but does not fully meet PLOS ONE’s publication criteria as it currently stands. Therefore, we invite you to submit a revised version of the manuscript that addresses the points raised during the review process.

We look forward to receiving your revised manuscript.

Kind regards,

Sreeram V. Ramagopalan

Academic Editor

PLOS ONE

Reviewers' comments:

Reviewer's Responses to Questions

**Comments to the Author**

1. Is the manuscript technically sound, and do the data support the conclusions?

Reviewer #1: Yes

Reviewer #2: Yes

2. Has the statistical analysis been performed appropriately and rigorously? 

Reviewer #1: Yes

Reviewer #2: Yes

3. Have the authors made all data underlying the findings in their manuscript fully available?

Reviewer #1: Yes

Reviewer #2: Yes

4. Is the manuscript presented in an intelligible fashion and written in standard English?

Reviewer #1: Yes

Reviewer #2: Yes

5. Review Comments to the Author

Reviewer #1: The following sentence may need some additional attention; if indeed EBV infection is present in all MS patients, this can be designated as a 'necessary factor' in the MS ethology. Whether or not the authors are willing to postulate this is important here, if the authors do not state this, this sentence needs rewriting and toning down.

"In addition, a prior Epstein Barr viral (EBV) infection seems to be a prerequisite for most (or all) genotypes in (G) to develop MS [3,4,49,50,60-62]. Indeed, if (as suggested by these studies) a prior EBV infection occurs in 100% of MS cases, this would indicate that EBV exposure is part of the causal pathway leading to MS and that, at least, this environmental exposure is required for disease pathogenesis [49]."

Ellers minor comments:

For the sake of understanding where the number 0.99 comes from please change the sentence:

“"For example, in HIV, if homozygous Δ-32 mutations were completely protective, then: P(G) = 0.99 .

To

"For example, in HIV, if homozygous Δ-32 mutations (occurring in 1% of the population) were completely protective, then: P(G) = 0.99 .”

And similarly

"By contrast, in SCD, where: P(G) = 0.03 , we would characterize carrying homozygous HbS mutations as the defining trait for membership in the “genetically susceptible” subset."

To

"By contrast, in SCD, where: P(G) = 0.03 , we would characterize carrying homozygous HbS mutations (3% of individuals) as the defining trait for membership in the “genetically susceptible” subset."

Discussion:

The authors state that at tiny fraction of the population is genetically susceptible and refer to the number of less than 4,7%; This is not a ‘tiny’ fraction, and could rather be described as a merely ‘fraction’

Reviewer #2: The paper is an interesting look at GWAS data, prevalence data, sibling and twin studies, and changes in the sex ratio of MS over time. The female to male sex ratio has been increasing over time, and the incidence and prevalence of MS has generally increased over time. The reasons for these changes are unknown.

For readers who are not familiar with IMSGC study (Science 2019, 365:eeav7188), rather than “over 200 genes” in the Abstract and Introduction, it might be worth saying that based on SNP data there are currently 233 genes associated with MS susceptibility, including 32 genes within the MHC, and the first locus identified on a sex chromosome, on the X chromosome. The SNPs are located within or near to immune related genes, and implicate both the adaptive and innate arms of the immune system. The MHC DRB1*15:01-DQB1*06:02 haplotype has the strongest association, with about a 3-fold increased risk of MS. About 23% of the general population in Europe and North America carries this haplotype, but 80% of this group are not at risk of MS.

Regarding life expectancy of people with MS, p.15 and references #37-41 dating from 2008-2014, I think it is worth pointing out that life expectancy has been improving, e.g. Koch-Henriksen N, et al. J Neurol Neurosurg Psychiatry 2017;88:626–631.

The prevalence data cited on p.16, references #42 and #43 date from 1997 to 2001. There is more recent data from Wallin MT et al 2019, Neurology March 5, 92(10): e1-e12, which estimates the prevalence of MS in the US fat between 337.9 per 100,000 population (n = 851,749 persons with MS), to 362.6 per 100,000 population (n = 913,925 persons with MS).

The reference cited in the legend to Fig 3, I think should be #54 Orton et al 2006, rather than reference #58.

There are several interesting conclusions in the paper. It appears that only about 4.7% of the population in Europe and North America is susceptible to ever developing MS. I think the statement that “MS is fundamentally a genetic disorder” is perhaps too strong. The point being made is that only a small proportion of the population is at risk for developing MS. In the Introduction and Discussion, the authors note that susceptibility to MS involves both environmental and genetic factors. There are few diseases which are purely genetic or purely environmental, and there is an interaction between genes and environment to varying degrees. MS cannot develop without the right environmental exposures. The paper takes the position that genetics makes the predominant contribution.

Based on twin studies and migration studies, in the Discussion the authors note that two or more environmental events probably contribute. One environmental exposure occurs during early life in the intrauterine or early postnatal period, and a second event occurs sometime before the age of about 15 years. EBV infection and vitamin D deficiency appear to be important as well. Even in someone with a susceptible genetic background and the correct environmental exposure, more than 50% will still not develop the disease, indicating a stochastic element.

Another interesting conclusion is that “men are more likely than women to be genetically susceptible to MS”. Men are 2-4 x more likely to be in the genetically susceptible subset, but disease penetrance is less. This is counter intuitive as the observed female:male ratio is about 3:1, and there is an MS associated SNP on the X chromosome, as noted.

Regarding the missing heritability in GWAS studies, I think the reasons given in the last paragraph on p.36 as an alternative explanation are more likely. The genes identified through GWAS studies explain about 48% of the estimated heritability for MS. GWAS studies cannot detect rare mutations, copy number variants, epigenetic effects, etc.

Dr. Goodin is a highly respected MS expert and has made significant contributions to the field. I think the paper makes a good addition to the literature around the genetics of autoimmune and other complex diseases.

The paper is not an easy read. There is a fairly long section in the middle in which it is possible to become lost in the mathematical symbols, and the need for reference back to the symbols or to Table 1. I think this detracts from the flow of the paper and the key ideas proposed. Perhaps more of the statistical and mathematical treatment could be in the supplementary section. The formula with an explanation in words in the text, with the derivation of the formula in a supplementary section or footnote, might be easier to follow.

6. PLOS authors have the option to publish the peer review history of their article (what does this mean?). If published, this will include your full peer review and any attached files.

Reviewer #1: No

Reviewer #2: No

---

## [Author Response · Author response to Decision Letter 0]

10 Jan 2021

Dear Dr. Ramagopalan:

Re: The Nature of Genetic and Environmental Susceptibility to Multiple Sclerosis

Goodin DS, Khankhanian P, Gourraud PA, Vince N

PONE-D-20-25332

Thank you very much for your letter of 30 December 2020 regarding the above referenced manuscript. Enclosed please find a new version of this manuscript, which has been revised in accordance with the Reviewers comments. I hope that, with this revision, you will now find the manuscript suitable for publication in PLoS One. Specifically, we have made the following changes to address the Reviewers concerns:

Reviewer #1: 

1. The following sentence may need some additional attention; if indeed EBV infection is present in all MS patients, this can be designated as a 'necessary factor' in the MS ethology. Whether or not the authors are willing to postulate this is important here, if the authors do not state this, this sentence needs rewriting and toning down.

"In addition, a prior Epstein Barr viral (EBV) infection seems to be a prerequisite for most (or all) genotypes in (G) to develop MS [3,4,49,50,60-62]. Indeed, if (as suggested by these studies) a prior EBV infection occurs in 100% of MS cases, this would indicate that EBV exposure is part of the causal pathway leading to MS and that, at least, this environmental exposure is required for disease pathogenesis [49]."

Response: As requested, we have now indicated that, in this circumstance, EBV infection can be designated as a ‘necessary factor’ (p. 26).

2. For the sake of understanding where the number 0.99 comes from please change the sentence: “"For example, in HIV, if homozygous Δ-32 mutations were completely protective, then: P(G) = 0.99 . 

To: "For example, in HIV, if homozygous Δ-32 mutations (occurring in 1% of the population) were completely protective, then: P(G) = 0.99 .”

And similarly

"By contrast, in SCD, where: P(G) = 0.03 , we would characterize carrying homozygous HbS mutations as the defining trait for membership in the “genetically susceptible” subset."

To "By contrast, in SCD, where: P(G) = 0.03 , we would characterize carrying homozygous HbS mutations (3% of individuals) as the defining trait for membership in the “genetically susceptible” subset."

Response: We have now modified these sentences as suggested (p. 10)

3. The authors state that at tiny fraction of the population is genetically susceptible and refer to the number of less than 4,7%; This is not a ‘tiny’ fraction, and could rather be described as a merely ‘fraction’

Response: We have now modified this sentence as suggested (p. 35)

Reviewer #2: 

1. For readers who are not familiar with IMSGC study (Science 2019, 365:eeav7188), rather than “over 200 genes” in the Abstract and Introduction, it might be worth saying that based on SNP data there are currently 233 genes associated with MS susceptibility, including 32 genes within the MHC, and the first locus identified on a sex chromosome, on the X chromosome. The SNPs are located within or near to immune related genes, and implicate both the adaptive and innate arms of the immune system. The MHC DRB1*15:01-DQB1*06:02 haplotype has the strongest association, with about a 3-fold increased risk of MS. About 23% of the general population in Europe and North America carries this haplotype, but 80% of this group are not at risk of MS.

Response: As suggested, we have now incorporated these important points into our Abstract and Introduction (p.3 & p.6). 

2. Regarding life expectancy of people with MS, p.15 and references #37-41 dating from 2008-2014, I think it is worth pointing out that life expectancy has been improving, e.g. Koch-Henriksen N, et al. J Neurol Neurosurg Psychiatry 2017;88:626–631.

Response: We agree and have now referenced this paper and added a comment regarding these findings (p.11).

3. The prevalence data cited on p.16, references #42 and #43 date from 1997 to 2001. There is more recent data from Wallin MT et al 2019, Neurology March 5, 92(10): e1-e12, which estimates the prevalence of MS in the US fat between 337.9 per 100,000 population (n = 851,749 persons with MS), to 362.6 per 100,000 population (n = 913,925 persons with MS).

Response: We agree that this is an important study needs to be referenced. We have now included a reference to, and a brief discussion of, their findings (p.11).

4. The reference cited in the legend to Fig 3, I think should be #54 Orton et al 2006, rather than reference #58.

Response: The Reviewer is correct. This has now been changed (Fig 3).

5. I think the statement that “MS is fundamentally a genetic disorder” is perhaps too strong. The point being made is that only a small proportion of the population is at risk for developing MS. In the Introduction and Discussion, the authors note that susceptibility to MS involves both environmental and genetic factors. There are few diseases which are purely genetic or purely environmental, and there is an interaction between genes and environment to varying degrees. MS cannot develop without the right environmental exposures. The paper takes the position that genetics makes the predominant contribution.

Response: We agree completely with the Reviewer. Our statement incorrectly implied that genetics was of predominant importance and we have now modified it to make the point that both genetics and the environment are required (p.35).

6. The paper is not an easy read. There is a fairly long section in the middle in which it is possible to become lost in the mathematical symbols, and the need for reference back to the symbols or to Table 1. I think this detracts from the flow of the paper and the key ideas proposed. Perhaps more of the statistical and mathematical treatment could be in the supplementary section. The formula with an explanation in words in the text, with the derivation of the formula in a supplementary section or footnote, might be easier to follow.

Response: We appreciate the fact that this paper is somewhat difficult to read. We have now been over the manuscript and tried to make it more clear. We have also relaxed the requirement that, when the penetrance values of subsets (G1) and (G2) are different, the distribution of each are unimodal. This change has slightly increased our upper estimate of P(G) for Lower Solutions. We have also reduced the mathematics in the Main Text and moved some of this to the Supplemental Material. However, I suspect that the Reviewer is referring mostly to our presentations both of Proposition #1 and also of the environmental impacts on MS pathogenesis (Section #7). We feel that these particular sections are so central to our conclusions that we are very reluctant to move them. Nevertheless, we leave this to your editorial discretion.

I hope that, with these modifications and additions, you will now find the manuscript suitable for publication in PLoS One. Thank you very much for your consideration of this matter. I look forward to hearing from you in due course.

 Yours sincerely,

 Douglas S. Goodin, MD

 Professor of Neurology

 University of California, San Francisco

---

## [Editor Report · Decision Letter 1]

15 Jan 2021

The Nature of Genetic and Environmental Susceptibility to Multiple Sclerosis

PONE-D-20-25332R1

Dear Dr. Goodin,

We’re pleased to inform you that your manuscript has been judged scientifically suitable for publication and will be formally accepted for publication once it meets all outstanding technical requirements.

Kind regards,

Sreeram V. Ramagopalan

Academic Editor

PLOS ONE
---

## [Editor Report · Acceptance letter]

21 Jan 2021

PONE-D-20-25332R1 

The Nature of Genetic and Environmental Susceptibility to Multiple Sclerosis 

Dear Dr. Goodin:

I'm pleased to inform you that your manuscript has been deemed suitable for publication in PLOS ONE. Congratulations! Your manuscript is now with our production department. 

Kind regards, 

on behalf of

Dr. Sreeram V. Ramagopalan 

Academic Editor

PLOS ONE